# The Road to Safety: A Review of Uncertainty and Applications to Autonomous Driving Perception

**DOI:** 10.3390/e26080634

**Published:** 2024-07-26

**Authors:** Bernardo Araújo, João F. Teixeira, Joaquim Fonseca, Ricardo Cerqueira, Sofia C. Beco

**Affiliations:** Bosch Car Multimedia S.A., 4705-820 Braga, Portugal; bernardo.araujo@pt.bosch.com (B.A.); joaquimlrangel@gmail.com (J.F.); ricardo.cerqueira@pt.bosch.com (R.C.); sofia.beco@pt.bosch.com (S.C.B.)

**Keywords:** deep learning, safety, autonomous driving, uncertainty quantification, calibration, out-of-distribution detection, active learning

## Abstract

Deep learning approaches have been gaining importance in several applications. However, the widespread use of these methods in safety-critical domains, such as Autonomous Driving, is still dependent on their reliability and trustworthiness. The goal of this paper is to provide a review of deep learning-based uncertainty methods and their applications to support perception tasks for Autonomous Driving. We detail significant Uncertainty Quantification and calibration methods, and their contributions and limitations, as well as important metrics and concepts. We present an overview of the state of the art of out-of-distribution detection and active learning, where uncertainty estimates are commonly applied. We show how these methods have been applied in the automotive context, providing a comprehensive analysis of reliable AI for Autonomous Driving. Finally, challenges and opportunities for future work are discussed for each topic.

## 1. Introduction

In a safety-critical domain, such as perception in Autonomous Driving (AD), it is crucial to mitigate safety and reliability concerns, such as the ones identified in [1]. Despite their high complexity and generalisation capability, Deep Neural Networks present safety flaws, as they natively have a black-box behaviour, tend to give overconfident predictions, and fail in situations for which they were not trained. On the other hand, depending on the training methodologies applied, these models can be improperly led to overfit the training data, not sufficiently covering the Operational Domain or failing in rare or difficult scenarios. Self-driving vehicle models must make near-instantaneous, safety-critical decisions with limited data, needing a swift and precise evaluation of the uncertainty in their environment perception.

In this review, we focus on the problems of Uncertainty Quantification (UQ), model calibration, out-of-distribution (OOD) detection and active learning (AL) in the context of improving Artificial Intelligence (AI) safety, specifically in the domain of perception in AD.

In this context, we aim to provide an answer to the following two research questions:What are the main UQ methods proposed across the literature, possible method categories and differentiating features?How can and has UQ been used to improve the reliability of perception models in AD?

We first explore existing UQ approaches that measure the lack of confidence in the prediction, with randomness and novelty as the main sources of error. This is often used as the basis for other safety methods.

Furthermore, calibration aims to reduce overconfidence by setting the confidence closer to the probability of the event. For example, the classification of an object as a car with a confidence of 80% should be correct 80% of the time. Uncertainty estimates can be used for OOD, making perception models more suited to open-world contexts, where unknown object classes and unexpected distribution shifts are possible. On the other hand, uncertainty can also be used to signal samples to label in an AL process.

These safety methods can be applied to AD perception problems, which include different tasks such as classification, object detection and semantic segmentation, and different modalities of data, such as RGB and Light Detection and Ranging (LiDAR). To varying degrees, these topics have been the subject of the AD applied research and, therefore, we complement the overview with other works that can be adapted to this field in the future or serve as a reference to new approaches. We also present some fundamental definitions and metrics for the comprehension of these subjects. In summary, the main contributions of this paper are as follows:Offer a collection and description of the relevant State-of-the-Art (SOTA) about safe approaches concerning perception algorithms and their application in AD.Analyse the SOTA in the AD field, pointing out current gaps, the most relevant findings and future challenges.Give a global overview of several interconnected topics of safe AI that are important to AD.

## 2. Uncertainty Quantification

Perception models, like all deep learning models, can make mistakes. In a safety-critical domain such as AD, it is essential to have an understanding of when perception models are likely to be wrong. This can be estimated by quantifying their uncertainty and communicating it to the user or system, depending on the level of automation (1 to 5), where the maximum level corresponds to full driving automation. This section will introduce the SOTA for Uncertainty Quantification (UQ) by analysing the multiple definitions of uncertainty that exist in the literature, the most important methodologies, and finally how they are implemented in the field of AD.

We categorise the different definitions of uncertainty in the literature into predictive uncertainty and distance-based uncertainty in Section 2.1 and Section 2.2, respectively, explaining their concept and implications regarding model architecture, training, and inference.

### 2.1. Predictive Uncertainty

There are two main sources of error: randomness and divergence from the training set. The predictive uncertainty concept allows us to quantify these sources of error with aleatoric and epistemic uncertainties, respectively. Decomposing uncertainty into these two parts allows taking corrective measures specifically tailored to each type of uncertainty, improving the safety of the overall system.

In a Bayesian framework, predictions of target variables can be obtained by:(1)$$p\left(y|x,D\right)=E_{p(\theta |D)}\left[p\left(y|x,\theta \right)\right],$$
where *y* is the target variable, *x* is the input sample, *D* is the dataset, and θ is the set of model parameters. The total uncertainty of a given prediction can be defined as the entropy of the predicted distribution of the target variable [2]:(2)$$\mathrm{Total}\mathrm{Uncertainty}=H\left[p(y|x,D)\right]=H\left[E_{p(\theta |D)}\left[p(y|x,\theta )\right]\right],$$
where H[p(x)] refers to the entropy of distribution p(x). As such, the total uncertainty can be decomposed into two components, aleatoric and epistemic [3]:(3)TotalUncertainty=AleatoricUncertainty+EpistemicUncertainty.
Aleatoric uncertainty [2] can be defined as:(4)$$\mathrm{Aleatoric}\mathrm{Uncertainty}=E_{p(\theta |D)}\left[H\left[p(y|x,\theta )\right]\right],$$
and epistemic uncertainty [2] as:(5)EpistemicUncertainty=TotalUncertainty−AleatoricUncertainty.

UQ is the problem of computing these values for specific samples and models. Uncertainties are typically measured in units of information such as bits or nats. To perform UQ and decompose it into its aleatoric and epistemic components, both p(y|x,θ) and p(θ|D) must be modelled.

Quantifying aleatoric uncertainty requires modelling p(y|x,θ), which involves having a model able to assign a probability to each value in the domain of the target variable. Most SOTA classification models already do this. In regression, however, most SOTA models must be modified to support this. In Section 2.3.1, we review the most prominent methods of adding this capability to current Deep Learning (DL) models.

Quantifying epistemic uncertainty requires obtaining a distribution of model parameters (Figure 1), which is extremely challenging due to the complex nature of this distribution. In the Bayesian framework, this distribution, p(θ|D), is calculated using Bayes’ rule:(6)$$p\left(\theta |D\right)=\frac{p(D|\theta )p(\theta )}{p(D)}.$$
where p(θ|D) is called the posterior, p(D|θ) is called the likelihood, p(θ) is called the prior and p(D) is called the evidence. Bayesian neural networks (BNNs) are neural networks where a distribution over parameters p(θ|D) is computed using this expression [4]. p(θ|D) is typically intractable to compute exactly and so it must be approximated. This is called a variational approximation, and can be denoted by $q_{\theta }\left(w\right)$, where *w* are weights of the original network architecture and θ are the variational parameters. See [5] for more details. This distribution over parameters can then be used to obtain predictions by applying Equation (Equation 1). In Section 2.3.2, we review some of the most promising methods of quantifying epistemic uncertainty in DL models. As noted by Abdar et al. [4], most works in UQ use this definition of uncertainty or a related one.

Related with uncertainty is the notion of risk, which corresponds to the total expected loss. Jain et al. [8] define the risk as:(7)$$\mathrm{Risk}\left(f\right)=E_{P(X,Y)}\left[\ell (Y,f(X))\right],$$
where *ℓ* represents a loss function of a model that learns to predict, and the point-wise risk or the expected loss is:(8)$$\mathrm{Risk}\left(f,x\right)=E_{P(Y|X=x)}\left[\ell (Y,f(x))\right].$$
The Bayes-optimal predictor $f^{*}$ is the model that minimises the risk:(9)$$f^{*}\left(x\right)=\underset{\tilde{y}}{\mathrm{argmin}}E_{p(Y|X=x)}\left[\ell (Y,\tilde{y})\right].$$
Bayes risk is the risk of the Bayes-optimal predictor [9]:(10)$${\mathrm{Risk}}_{\mathrm{Bayes}}\left(x\right)=\mathrm{Risk}\left(f^{*},x\right).$$
The point-wise risk can be decomposed into the Bayes risk and the Excess risk:(11)$$\mathrm{Risk}(f,x)={\mathrm{Risk}}_{\mathrm{Bayes}}(x)+{\mathrm{Risk}}_{\mathrm{Exc}}(f,x).$$

Some authors define the total uncertainty as the point-wise risk, aleatoric uncertainty as the Bayes risk, and epistemic uncertainty as the Excess risk [8,9,10,11]. This risk definition of aleatoric uncertainty implies that it is irreducible, or inherent to the data, and so it is model-independent. It should be noted, however, that in practice, with this definition, it is not possible to compute the aleatoric uncertainty, and consequently also the epistemic uncertainty since the Bayes-optimal model is unknown. The total uncertainty can be computed but as an approximation since we only observe a single sample from the distribution of y and not the distribution itself.

This uncertainty definition is incompatible with the definitions of uncertainty in Equations (Equation 2), (Equation 4) and (Equation 5). In the latter, the aleatoric uncertainty is the uncertainty which is presumed by the model to be irreducible, and so it is model-dependent [3]. Uncertainties in this latter definition can also be interpreted as estimates of the risk and its components [12]. Nonetheless, there is some inconsistency with the concept of aleatoric uncertainty since some authors [13,14,15] define it as irreducible with more data while using the mathematical definitions of Equations (Equation 2), (Equation 4) and (Equation 5), corresponding to the model-dependent uncertainty definition.

### 2.2. Distance-Based Uncertainty

Another definition of uncertainty is the distance, in some space, between a sample and the training dataset. For example, Liu et al. [16] define uncertainty as:(12)$$\begin{array}{cc} u(x) & =v\left(d(x,X_{IND})\right),\\ \;d(x,X_{IND}) & =E_{x^{\prime }∼X_{IND}}||x-x^{\prime }{\left|\right|}_{X}^{2}, \end{array}$$
where u(x) quantifies model uncertainty, *v* is a monotonic function, *x* is the input sample, $X_{IND}$ is the set of training samples, and $d(x,X_{IND})$ is the expected distance between sample *x* and a sample in the training set, according to some semantically meaningful distance metric $\left|\right|\cdot {\left|\right|}_{X}$.

Deterministic Uncertainty Quantification (DUQ) groups the samples belonging to a single class into clusters and summarises them into a centroid, which is updated using an exponential moving average of the embedding of the samples throughout training (see Figure 2). It then quantifies uncertainty by measuring the distance of a sample to its closest centroid in this embedding space [17]. Distance-based uncertainty methods explicitly consider the relationship between data samples, whereas other types of epistemic Uncertainty Quantification methods must implicitly encode these relationships. This may make the concept of distance-based uncertainty more intuitive than entropy-based uncertainty. However, distance in feature space may not necessarily reflect the differences in the loss function, which may be a desirable property.

### 2.3. Methods

This section focuses on the most relevant literature on quantifying both aleatoric and epistemic uncertainty, with most publications adopting the predictive uncertainty definition. Methods with generic application are first described, while AD applications are mostly explored towards the end of the section. Table 1 lists the cited papers.

#### 2.3.1. Aleatoric Uncertainty Quantification

Assuming the definition of uncertainty as a prediction of the risk, aleatoric uncertainty is the uncertainty which is considered to be irreducible by a model, and is related to the inability of a model to predict the exact value of a target variable within a distribution. Aleatoric uncertainty in AD can be caused by sensor noise, object occlusion, and distance to the sensor, as well as challenging environmental conditions such as rain, fog, snow, sensor blockage and interference, which usually degrade the model’s performance [55,56,57]. Therefore, aleatoric UQ can serve a crucial role in identifying these potentially dangerous factors, which would allow the autonomous vehicle to implement defensive behaviours in response, such as bringing it to a safer speed or fully stopping the vehicle in a safe area.

One family of methods aimed at quantifying aleatoric uncertainty is *direct modelling*. This consists in directly estimating the parameters of a chosen distribution for the output of a model. Direct modelling was first proposed in Nix and Weigend [58] and used to decompose epistemic and aleatoric uncertainties by Kendall and Gal [18]. For example, in regression, one possible choice for the distribution of the model output is the Gaussian distribution which has parameters μ and $\sigma^{2}$. The task then is to incorporate these parameters into the model and loss function.

Assuming heteroscedastic uncertainty, i.e., uncertainty that varies with the input, the parameter $\sigma^{2}$ of the Gaussian distribution is an output of the model. These parameters can be integrated into the loss function as follows:(13)$$L\left(\theta \right)=\frac{1}{N}\sum_{i=1}^{N}\left[\frac{1}{2\sigma_{\theta }{\left(x_{i}\right)}^{2}}{∥y_{i}-f_{\theta }\left(x_{i}\right)∥}^{2}+\frac{1}{2}\log\sigma_{\theta }{\left(x_{i}\right)}^{2}\right].$$

For homoscedastic uncertainty, the uncertainty does not vary with respect to the input, and therefore the parameter $\sigma^{2}$ is constant. Furthermore, as it changes the loss function, the direct modelling of aleatoric uncertainty can lead to more accurate predictions by penalising the model less in noisy samples. The authors registered an improvement of 1–3% on the Make3D [59], CamVid [60] and NYUv2 [61] datasets for the tasks of semantic segmentation and depth regression.

One approach to build lightweight probabilistic networks is presented by Gast and Roth [19], where the activations of intermediate layers are modelled as normal distributions, leaving the weights as point estimates. At each layer, they compute the parameters of the Gaussian distribution which minimise the Kullback–Leibler (KL) divergence with the distribution of the activations, which is equivalent to computing their mean and covariance. Probabilistic output layers are also used, enabling the quantification of aleatoric uncertainty: a general power exponential layer for regression problems, and a Dirichlet layer for classification.

Wang et al. [20] employ a different process for aleatoric uncertainty estimation using Test-Time Augmentation (TTA). They start by modelling the augmentation transformations applied to the input images, considering them to be part of the image acquisition process, represented by hidden parameters. For this, the authors use spatial transformations, namely, rotation, scaling, and flipping, as well as intensity noise. Using these multiple augmented images at test time reportedly enables the capture of aleatoric variability. The model then produces multiple predictions for the same image, each being perturbed by a differently parameterised augmentation transformation, approximating the expected value of the output *Y* given the perturbed input *X*, via Monte Carlo (MC) simulation. The authors use these predictions to estimate the aleatoric uncertainty by computing the entropy related to the image transformations and noise, and average over multiple sets of transformation parameters. The pixel-wise uncertainty estimation by MC samples by test-time dropout is then extended for entire structures/lesions by measuring the variation in volume between segmentation predictions of different MC samples. The structure-wise uncertainty is estimated using the volume variation coefficient (VVC), which uses the mean $\mu_{\nu }$ and standard deviation $\sigma_{\nu }$ of the set of volumes ν:(14)$$\mathrm{VVC}=\frac{\sigma_{\nu }}{\mu_{\nu }}.$$

Older TTA applications can be found in [62,63,64].

#### 2.3.2. Epistemic Uncertainty Quantification

Epistemic uncertainty comes from a lack of knowledge about the process underlying the target variable. In the case of neural networks, this lack of knowledge originates from an insufficient coverage of the Operational Domain (OD) by the training set. Samples which are sufficiently different from the training set present a great challenge to perception models such as neural networks, as they may be unable to generalise to these cases. In fact, they have been shown to make confidently incorrect predictions in these types of cases [65,66].

This type of uncertainty can have different purposes depending on where in the perception pipeline it is applied. In a development or other offline environment, it can be used to select informative samples for labelling that differ from the existing training set in a model-relevant way. This process, known as active learning (AL), can reduce the epistemic uncertainty of models trained with the newly labelled examples as well as increasing their performance. A more detailed explanation of active learning is given in Section 4.2.

On the other hand, when deployed in real-world systems, this type of uncertainty can allow the perception models to identify new types of objects which are not present or labelled in the training dataset, a process known as novelty or OOD detection. Otherwise, if left unidentified, these could induce incorrect, unsafe or unpredictable behaviours in the Autonomous Vehicle (AV), limiting their safety.

Several alternatives have been proposed throughout the literature in order to model the posterior weight distribution of neural networks. This allows for the computation of epistemic uncertainty, i.e., model uncertainty or parameter uncertainty. Some of the most relevant works are listed in this section.

Ensembles are sets of models used in conjunction to produce a single prediction, typically resulting in better performance in many tasks [67], and simultaneously allowing for epistemic UQ. The first use of ensembles to produce uncertainty estimates is demonstrated in Lakshminarayanan et al. [21]. In this work, multiple models are trained independently using the entire dataset but starting from different initialisation. The use of bagging, which consists in training each model with a different subset of the data, is found to decrease performance. Direct modelling is used to represent aleatoric uncertainty. The prediction of the ensemble is the average of the predictions of the models:(15)$$p_{E}\left(y|x\right)=\frac{1}{M}\sum_{m}^{M}p_{\theta_{m}}\left(y|x\right),$$
where $p_{E}\left(y|x\right)$ is the prediction of the ensemble, $p_{\theta_{m}}\left(y|x\right)$ is the prediction of a single model of the ensemble, and *M* is the number of models in the ensemble. Analogously to epistemic uncertainty, the authors compute a disagreement measure, given by:(16)$$\mathrm{Disagreement}=\sum_{m=1}^{M}D_{KL}\left(p_{\theta_{m}}\left(y|x\right)\left|\right|p_{E}\left(y|x\right)\right),$$
where $D_{KL}\left(p\left(x\right)||q\left(x\right)\right)$ is the KL divergence. The authors show that this disagreement method is useful for distinguishing between known and unknown classes and that its performance increases for increasing ensemble sizes *M*. Different works have shown that ensembles often outperform other epistemic UQ methods in a variety of different tasks and datasets [22,49,68].

Another set of methodologies to estimate epistemic uncertainty is HydraNets. First proposed by Lee et al. [69] as TreeNets, HydraNets are a family of network architectures with multiple outputs for a single target variable. Lee et al. [69] introduce them as approximations to ensembles where a subset of the parameters is shared (see Figure 3). Specifically tailored objective functions based on Multiple Choice Learning (MCL) are used to promote diversity within the ensembles, which can be trained end-to-end and result in improved accuracies versus traditional ensembles.

For UQ, the use of these shared-parameter schemes was first explored by Osband et al. [70]. The authors use a model with multiple heads and a shared network for Reinforcement Learning (RL) through the proposed Bootstrapped Deep Q Learning methodology. The results show that the use of HydraNet-structured models results in more efficient exploration in several RL tasks. Addressing the task of visual odometry, Peretroukhin et al. [23] apply HydraNets to predict rotations. They compute both aleatoric and epistemic uncertainties (see Figure 4) using a DL HydraNet-based model and classical odometry, attaining more robust fused estimates on the KITTI [71] and 7-Scenes [72] datasets. With HydraNet’s multi-headed structure, the authors aim to model epistemic uncertainty when the model’s inputs differ from those seen during training. Overall, the multi-headed structure and shared parameters of HydraNets have shown promise in efficiently and effectively modelling uncertainty. As shown by the works presented above, this is demonstrably true in topics such as RL and visual odometry, closely related to the field of AD in general.

Furthermore, the HydraNet architecture can help overcome one disadvantage of using full-fledged ensembles for UQ: the need to train multiple models and execute them at inference time, resulting in high computational requirements, impractical for real-time applications like AD. Knowledge distillation of an ensemble is a possible approach to compress the set of models into a smaller and faster architecture, for deployment purposes. However, standard knowledge distillation methods [73] only return a single aggregate of the various predictions of the ensemble, generally the average. In order to compute uncertainties, it is important to keep the individual predictions and, as such, Hydra Distillation by Tran et al. [24] can solve this issue. This method assigns a model from the teacher ensemble to a head of the student HydraNet model and, through the minimisation of the KL divergence between each pair’s outputs, it trains the student’s heads and singular shared backbone, hence resulting in a smaller and more efficient model. From these outputs, it is possible to compute the total, aleatoric and epistemic uncertainties. Experiments on classification and regression datasets showed improved accuracy, Negative Log Likelihood (NLL) and Brier Score (BS) compared to previous distillation methods.

Employing a different approach from ensembles, MC Dropout, proposed by Gal and Ghahramani [25], is a widely adopted methodology for UQ due to its flexibility and simplicity of implementation. Dropout [74] is a common regularisation technique which randomly sets activations to zero with some probability *p*. Applying this technique at test time, a distribution over the weights can be obtained, commonly named the Dropout Distribution. These samples can then be drawn during testing, producing a distribution of the outputs in relation to the parameter distribution. From these distributions, both aleatoric and epistemic uncertainties can be computed. Due to the simplicity and high flexibility of this method, it has been employed in a wide variety of applications, such as medical imaging analysis [75,76,77,78,79,80], AD [45,46,47,48], OOD detection [81], and AL [82], to name a few.

Kingma et al. [26] show that a variant of dropout known as Gaussian Dropout [83], where Gaussian, rather than Bernoulli noise, is applied to the hidden units, can also be used for uncertainty estimation, while the associated dropout probability is learned automatically, resulting in better performance. McClure and Kriegeskorte [27] find that stochastic regularisation methods such as DropConnect [84,85], Gaussian DropConnect, Gaussian Dropout [83], and their combinations can also be used to obtain uncertainty estimates, with improvements in performance. Figure 5 shows the differences between these regularisation methods.

Similarly to ensembles, one disadvantage of MC Dropout is the need to perform multiple forward passes at inference time. Some works have attempted to mitigate this problem by distilling the MC Dropout ensemble into a single student model [28] or designing an architecture to match the first and second moments of the MC Dropout ensemble [29].

An alternative method is Bayes by Backprop (BBB), proposed by Blundell et al. [7]. It is an effective, backpropagation-compatible variational method to estimate a posterior of the parameters of a neural network. Besides allowing the estimation of an accurate posterior, the method also automatically applies regularisation to the network. These effects are achieved through the minimisation of the compression cost. The so-called compression cost is equivalent to the variational free energy or the expected lower bound (ELBO) on the marginal likelihood. This makes the technique simple to implement and enables the modelling of more intractable distributions. BBB approximates the parameter posterior with a variational approximation:(17)$$q_{\phi }\left(\theta \right)\approx p\left(\theta |D\right),$$
which uses the mean field approximation, where the components $q_{\phi_{i}}\left(\theta_{i}\right)$ of the distribution are independent:(18)$$q_{\phi }\left(\theta \right)=\prod_{i=1}^{N}q_{\phi_{i}}\left(\theta_{i}\right).$$

In the case of BBB, the distribution $q_{\phi_{i}}\left(\theta_{i}\right)$ is the normal distribution:(19)$$q_{\phi_{i}}\left(\theta_{i}\right)=N\left(\mu_{i},\sigma_{i}^{2}\right).$$

To sample from this distribution in a way which enables the training of the parameters, the reparameterisation trick [26] is used. This involves first sampling ϵ∼N(0,I). A sample from the weight posterior can then be obtained by setting θ=μ+σ⊙ϵ. The parameters μ and σ can then be learned through backpropagation as they would normally. This is equivalent to learning a set of weights which are normally distributed, with a diagonal covariance matrix: θ∼N(μ,σI). BBB has since been extended to Recurrent Neural Networks [87], Convolutional Neural Networks (CNNs) [30,88] and applied to Continual Learning [31] and AL [89].

Another Bayesian method is presented by Ritter et al. [32], where UQ capabilities can be added to any trained network, without the need of a specific training procedure. The weight posterior is modelled as a Gaussian distribution around the (already computed) weight Maximum a posteriori (MAP) estimate, using a Laplace approximation of the loss curvature of the network. Several forward passes are then performed with sets of weights sampled from the approximate weight posterior. This method requires low memory usage and is computationally efficient. Classification tests performed with MNIST [90] and notMNIST [91] demonstrated better robustness to simple adversarial attacks and better out-of-distribution (OOD) detection than an MC Dropout approach. The computation of the Laplace approximation of the posterior over model parameters can, however, become intractable for very large models if a less restrictive (more expressive) approximation approach is chosen. To tackle this problem, subnetwork inference is proposed by Daxberger et al. [33]. The authors demonstrate that it is possible to apply the Bayesian treatment only to a subset of weights, and still preserve the predictive uncertainty. A strategy is devised in order to select which weights should become probabilistic, while the remaining weights are kept as their originally trained point estimates as Figure 6 suggests. Experiments with tabular regression and image classification under distribution shift show favourable results compared to ensembles and more restrictive posterior approximations over the complete model.

The Laplace family of approximations was further researched by Daxberger et al. [34] for the purpose of UQ. The authors claim the simplicity, performance, and low computational cost of this method, and further develop a public software library to enable DL practitioners to incorporate predictive uncertainty estimation into their models. Given a deterministic neural network, it is possible to define which weights should be probabilistic as pictured in Figure 7. An approximation of the Hessian should be chosen, depending on the intended precision–computation trade-off. Hyperparameter tuning follows, which can be performed during or after the standard training of the network. The method of extracting the probabilistic output can then be chosen.

The addition of uncertainty units (Learnable Uncertainty under Laplace Approximations (LULA) units) to the hidden layers of any neural network is proposed by Kristiadi et al. [35]. These units (see Figure 8) do not affect the final prediction but can be trained to improve the Laplace approximations’ performance in a decoupled manner from the standard training. They effectively change the geometry of the loss curvature in order to produce better calibrated uncertainties, after being trained via an uncertainty-aware loss on both In Distribution (ID) and OOD data. Experiments in OOD detection and prediction under dataset shift showed significantly better results with LULA units enabled, in terms of scoring and calibration, for classification tasks.

The work of Choi et al. [36] applies mixture density networks (MDNs) in order to perform uncertainty-aware regression in a sampling-free manner. This method outputs the parameters of a Gaussian Mixture Model (GMM). Aleatoric uncertainty is then computed as the average variance of the GMM, whereas epistemic uncertainty is the expected difference between the components of the GMM.

The application of MDNs for AL is later explored in [37]. Both epistemic and aleatoric uncertainties are computed for the object detection task and used to select informative samples to be labelled. For this effect, both the localisation and the classification heads of a conventional object detector are altered to predict the parameters of a GMM. Experiments on the PASCAL VOC [92] and MS-COCO [93] datasets show that this method achieves similar object detection performance compared to MC Dropout and ensemble-based uncertainty techniques, with lower computational cost.

A Bayesian hypernetwork is proposed by Krueger et al. [38] to model P(w|D). Hypernetworks are models that generate the weights for another network (the primary one). In this case, the Bayesian hypernetwork takes random noise as input and outputs a set of model parameters from the approximate posterior for the “primary” model, similarly to a generative model that samples a set of parameters (see Figure 9). Experiments are performed on MNIST, CIFAR10 [94] and a regression task for several use cases: regularisation, AL, anomaly detection, and the detection of adversarial examples. This approach is shown to be more robust to adversarial attacks than dropout.

On the other hand, the work of Singh and Principe [39] discards the Bayesian approach and uses concepts from quantum physics and perturbation theory to develop a new framework for deterministic single-shot UQ. Benchmarking on MNIST, K-MNIST [95], and CIFAR-10 shows that this approach delivers better results than Bayesian methods in several uncertainty quality metrics, including calibration, while being significantly faster.

Sankaranarayanan et al. [40] aim at providing uncertainties on the semantic latent variables, which has remained challenging despite recent advances in generative modelling and the subsequent representation of semantic information in disentangled latent spaces. The authors propose the estimation of principled uncertainty intervals which they guarantee contain true semantic factors for any underlying generative model. Their method relies on training an encoder (from StyleGAN2 [96]) on pairs of images to predict the semantic factors but also to estimate upper and lower conditional quantiles, effectively performing quantile regression for obtaining a heuristic uncertainty interval for each element of the latent space. A subsequent calibration step, employing rescaling factors, is undertaken to ensure that the naïve quantile interval contains the Ground Truth (GT) value in finite samples. Figure 10 presents an overview of the methodology and results.

Berry and Meger [41] provide a different way to estimate uncertainty. They employ Normalising Flows (NF) [97] in an ensemble fashion to reliably estimate epistemic uncertainty, while being flexible enough to still capture aleatoric uncertainty. They propose two versions: NF *ensemble out* provides an ensemble setting to the non-linear transformations *g*’s, in this case, cubic splines. For version two, the NF *ensemble base*, which is more focused on improving memory footprint, computational efficiency and error estimation, creates the ensemble in the base distribution, where the Gaussian parameters of mean, $\mu_{w}$, and variance, $\Sigma_{w}$, are estimated via a neural network with fixed dropout masks, using the following formulation:(20)$$\begin{array}{cc} p_{Y|X,W}\left(y|x,w\right) & =f_{\theta_{w}}\left(y,x\right)\\ \; & =p_{B|X,W}\left(g_{\theta }^{-1}\left(y,x\right)\right)\times \left|\det\left(J\left(g_{\theta }^{-1}\left(y,x\right)\right)\right)\right|, \end{array}$$
where $p_{B|X,W}\left(b|x,w\right)$ is the probability density function of $N(\mu_{w},\Sigma_{w})$. They further benchmark their uncertainty estimation approaches against MC Dropout and Probabilistic Network ensembles, both with fixed dropout masks, with each component modelling a Gaussian, and against a Gaussian Process. Evaluation is performed in five AL tasks, using the output of the K-Nearest Neighbours as GT, and averaging the KL divergence between the said GT and the estimates of 50 test samples. The authors report that NF ensemble models globally outperformed the remaining approaches, demonstrating their ability to estimate epistemic uncertainty and learning more expressive aleatoric uncertainty faster.

Another approach for UQ is using Evidential Deep Learning methods, in which, as output, the model produces the parameters of a distribution over distributions (evidential distribution). This distribution allows for the estimation of aleatoric and epistemic uncertainties without sampling in inference time, in contrast with sampling-based methods such as MC Dropout, making these methods more suitable for real-time applications. An important property of evidential methods is that, for unknown samples, the network can assume a uniform distribution, which implies total uncertainty. One of the first works to adopt this concept is from Sensoy et al. [42], who replace the standard softmax output activation function with ReLU and interpret this output as the parameter set of a Dirichlet distribution. The parameter set is fitted to best represent the predictions of the learner using a specific loss function. For a classification task, the results show that this method associates more uncertainty to out-of-distribution and wrong samples with adversarial perturbations, compared to other sampling-based methods, while keeping more accurate predictions with lower uncertainty.

Malinin and Gales [43] propose a similar approach named Prior Network. Besides quantifying model (epistemic) and data (aleatoric) uncertainties, the authors quantify the distributional uncertainty component, which is related to the mismatch between the training and test distributions. This is particularly relevant for the OOD detection task. A limitation of this method is the need to train the model with OOD data. This concept is also applied to regression by Amini et al. [44] in order to estimate aleatoric and epistemic uncertainties. For this task, high-order evidential priors are placed over the original low-order Gaussian likelihood function, and the evidential distribution hyperparameters are inferred by the neural network. This method has the advantage of not needing OOD data during training, while still showing increased uncertainty in OOD and adversarial data during test time. Further Evidential Deep Learning methods can be found in the survey of [98].

#### 2.3.3. Applications in Autonomous Driving

Feng et al. [45] suggest a network architecture for 3D object detection, performing tests with raw sequences from the KITTI dataset [99]. A region proposal network (RPN) is used, similar to the Faster-RCNN pipeline, to generate object proposals from a Bird’s-Eye View (BEV) format of the input point cloud. A sequence of three intermediate fully connected layers is defined, with dropout layers after each one.

To compute epistemic uncertainty, the dropout layers are enabled during training and inference time, to approximate a Bayesian model. This is the previously presented approach of MC Dropout. Multiple point estimates for the classification score are used to calculate the Shannon entropy of the prediction probability, the mutual information of the prediction probability, and the posterior of model parameters. Multiple estimates of the bounding box parameters are necessary to compute the total variance of the regression predictions, i.e., the trace of the covariance matrix of the normalised bounding box predictions.

Aleatoric uncertainty, on the other hand, does not require multiple passes with dropout enabled: a direct modelling approach is applied. An observation noise output vector is added to the network to predict the variance of the target variable. The regression loss is modified to promote higher values of this noise vector when the expected error is greater. The total variance of the vector is the aggregate of the variances of the target variable. The aleatoric uncertainty is not evaluated for the classification task.

The variation of epistemic uncertainty is shown to be associated with the detection accuracy, with greater values signalling samples different from the training dataset. Spatial aleatoric uncertainty, on the other hand, is dependent on the occlusion and the distance of an object from the ego vehicle. An improvement in detection performance by 1–5% is also reported when modelling the aleatoric uncertainty.

Regarding the RGB semantic segmentation task, a study is performed by Phan et al. [47], analysing the effects of different external influence factors on the perception performance and the uncertainty estimations of a Bayesian deep model. A synthetic dataset is used, ProcSy [100], previously developed by the authors. MC Dropout is used to approximate a Bayesian model, leveraging multiple inferences to compute the predictive entropy (total uncertainty), aleatoric entropy (aleatoric uncertainty), and mutual information (epistemic uncertainty). Depth, occlusion, rain, and the presence of clouds and puddles are introduced and varied in a synthetic AD dataset. It is found that, given enough data, epistemic uncertainty is reduced, but aleatoric uncertainty remains high for distant or occluded objects. Furthermore, clouds affect performance and uncertainty more negatively than rain or puddles. All types of uncertainty are proven to correlate to the segmentation performance, with the epistemic uncertainty having the highest correlation.

Regarding SOTA models for LiDAR semantic segmentation, SalsaNext is the first to include UQ in its implementation [48]. The authors propose a one-stage architecture capable of real-time inference with results ranking first in the SemanticKITTI [101] leaderboard at the time of publication. MC Dropout can be applied in order to compute epistemic uncertainties, while Assumed Density Filtering [19] enables the computation of aleatoric uncertainties.

Gustafsson et al. [49] have sought to compare the MC Dropout and ensembling approaches by evaluating the quality of the uncertainty measures and the scalability for real-time applications. Ensembles demonstrate better performance in terms of calibration and ability to rank predictions from least to most accurate based on the predictive uncertainty, using KITTI and Virtual KITTI [102] for depth regression, and Cityscapes [103] and Synscapes [104] for semantic segmentation.

Another use of UQ can be found in MonoFlex [50], a monocular 3D object detector that aims to decouple the detection of truncated objects from untruncated ones, using different approaches for each type. The depth of each object is estimated using four different methods, which are all supervised by losses that model the aleatoric uncertainty of the prediction. The final prediction is then formulated as the uncertainty-weighted average of these values. This soft ensemble is able to assign more weight to confident estimates while disregarding potentially inaccurate predictions. The direct modelling of uncertainties improves the depth estimation of all methods on the KITTI benchmark [71], and the implementation of the soft ensemble results in even better performance.

Catak et al. [51] propose the use of the area of the prediction surface as a metric for the epistemic uncertainty of image object detection models. PURE uses MC Dropout on object detection networks to infer predictions multiple times for each sample. The DBSCAN algorithm [105] clusters predictions based on the centre point of each of the resulting bounding boxes into objects. For each corner of each object’s bounding boxes, the convex hull is computed. The area of these surfaces is interpreted as the epistemic uncertainty of the model and is found to be correlated with the Intersection over Union (IoU) between the predicted and true bounding boxes. Evaluation is performed on KITTI, Stanford Cars [106], BDD100K [107], and NEXET [108] with three different models.

For the 2D object detection task, CertainNet [52] is proposed as a real-time model that estimates uncertainties for all output parameters, including size, location, class, and objectness. This work uses CenterNet as a base anchor-free detection architecture and builds upon the deterministic UQ ideas of DUQ [17] to adapt the model to infer sampling-free uncertainty. CertainNet does not require multiple inferences like Bayesian and ensemble methods (i.e., sampling-based approaches), nor does it explicitly model the uncertainty as an additional output. Instead, uncertainty-aware objectness, location and class are all predicted in a class heatmap for each pixel, while size is predicted in a dedicated heatmap. The objectness uncertainty is linked to the distance of the prediction encoding to the closest class centroid in the high-dimensional space. The centroid for each class is learned in the training process, similarly to DUQ. The class uncertainty, in turn, is given by the relative distance of the predicted embedding to the multiple class centroids. The location and size uncertainties are extracted from the variances of the predictions of those quantities made by the model in the areas around the object centre. Tests are performed with the KITTI, BDD100K, and nuImages [109] datasets.

GLENet [53] tries to compensate for the labelling issues that inevitably occur. LiDAR point cloud annotations are usually derived from 2D images and sparse point clouds. Similar objects, at similar distances, may be annotated with different estimated sizes, shapes and directions, especially when the objects are only partially visible. As such, the authors focus on improving the object detection network by employing the concept of location uncertainty estimation. For this, they first formulate the problem of 3D label uncertainty as a diversity of potentially plausible object bounding boxes (Figure 11). In essence, during inference, the generative network of GLENet, which is based on a Conditional Variational Auto-encoder (VAE), samples the latent space several times, to produce multiple possible detected bounding boxes to a prediction network downstream. Aided by conditional impositions of a context encoder, the result amounts to a bounding box prediction distribution, which encapsulates the label uncertainty. This work also proposes an uncertainty-aware quality estimator, based on fully connected layers, in order to facilitate the training of the localisation quality estimation branch and subsequent improvement of the IoU estimation accuracy. The KITTI and Waymo [110] datasets are used to demonstrate the effectiveness of GLENet. The overall workflow is depicted in Figure 12.

The work of Wu et al. [54] focuses on the applicability of UQ to empower a RL framework for AD. The interaction with the virtual environment (generated in the CARLA simulation platform [111]) is encoded onto an Ensemble Environment Model that predicts the transition dynamics of driving. The ensemble is specifically leveraged to compute the ensemble variance, and subsequently is able to estimate the epistemic uncertainty. This uncertainty is then used to adaptively determine the number of prediction rollout steps, truncating unreliable predictions. In particular, the higher the variance of the ensemble, the fewer rollout steps will be carried out.

It is apparent through this literature review that some very interesting approaches have already been proposed to apply UQ to AD perception tasks. The use of predictive, aleatoric, and epistemic uncertainty in semantic segmentation and object detection in both the RGB and the LiDAR domain are tackled in different works, with much room to improve and further develop. The usefulness of UQ for the AL use case has already been identified. Finally, the importance of model calibration in safety-critical problems like AD is already stated, with some work developed towards the recalibration of such models. Some of these publications very clearly state their weak points and limitations, providing important feedback and discussion regarding future work ideas.

## 3. Calibration

Deep Learning models that return probability distributions over the target domain are not necessarily calibrated, i.e., their confidence does not match the empirical probability of the prediction being correct. Thus, an assessment of the model’s calibration is required before safely interpreting its confidence as a probability. As such, multiple metrics can be used to evaluate a model’s calibration.

The property of calibration can only be evaluated if the model outputs a probability distribution. This is typically the case in classification problems (including semantic segmentation or the classification component of object detection), where the output typically consists in the parameters of a categorical distribution. However, in regression tasks (e.g., regression of the bounding boxes in object detection), an uncertainty modelling approach is required, where some measure of confidence is output by the model for each predicted parameter.

As with any safety-critical system, perception models in AD should be dependable and provide reliable prediction confidence. One way to promote this is by calibrating the model, a process that can also be called recalibration, typically with respect to a small calibration set. The calibration of Artificial Intelligence (AI) models concerns the improvement of the degree of confidence of their overall predictions. It would be intuitive to assume that a model that makes some prediction with a confidence score of 0.7 would, in fact, be correct in 70% of those predictions. However, this is generally not the case for learning models, as most trainings provide uncalibrated outcomes. Hence, it is relevant to study the ways that one can correct them.

Considering a binary classification problem, one way to tackle method calibration consists of learning to map the class-conditional density, also referred to as models’ scores, to empirical class membership probabilities (P(c|s(x)=s)). In this setting, a model can be said to be well calibrated if these estimates converge to the score value, for an arbitrarily large number of examples.

Table 2 summarises the calibration methods that are described in Section 3.2 and Section 3.3.

### 3.1. Evaluation Metrics

Models can quantify their uncertainty by predicting distributions over target variables. To evaluate the uncertainty estimation and, therefore, the calibration of these probabilistic models, it is insufficient to evaluate the correctness of their maximum confidence prediction as in most commonly used metrics (e.g., accuracy and IoU). As a consequence, there is the need for another set of metrics specifically to evaluate the model calibration [115].

#### 3.1.1. Calibration Error

In well-calibrated classification models, the confidence predicted by the model for a given class is equal to the empirical probability of the sample belonging to that class (e.g., a classification of an object as a car with a confidence of 80% should be correct 80% of the time). Similarly, in the case of regression models, the predicted confidence intervals at a given confidence level should include the true value with a probability equal to the confidence level (e.g., a confidence interval with a confidence level of 80% should include the true value 80% of the time). In addition, confidences of different models should have the same probabilistic interpretation so they are consistently interchangeable. Figure 13 shows the relationship between model confidence and the true probability in underconfident, overconfident, and calibrated models.

Calibration error metrics, such as Expected Calibration Error (ECE) and Maximum Calibration Error (MCE), measure the deviation between the model’s confidence and the empirical distribution of the data. To evaluate the calibration of a classification model, we can execute the following procedure [115]:Run inference over dataset and record model confidences.Split dataset into M bins according to model confidence (e.g., [0,0.1],⋯,[0.9,1.0]).For each bin $B_{m}$:(a)Compute the average accuracy of the model $acc(B_{m})$, the fraction of samples in the bin which are correctly classified;(b)Compute the average confidence of the model $conf(B_{m})$ for the positive class;(c)The calibration error for the bin is the absolute difference between model confidence and model accuracy $|acc\left(B_{m}\right)-conf\left(B_{m}\right)|$.

The top-label ECE, also known as confidence ECE, is then the average calibration error over every bin, weighted by the size of the bin [124,125]:(21)$${\mathrm{ECE}}^{\mathrm{top}-\mathrm{label}}=\sum_{m=1}^{M}\frac{\left|B_{m}\right|}{N}\left|acc\left(B_{m}\right)-conf\left(B_{m}\right)\right|,$$
where *N* is the number of samples, $B_{m}$ is the *m*-th bin, $acc(B_{m})$ is the accuracy of the model in the bin, and $conf(B_{m})$ is the average confidence of the model in the bin.

However, other formulations of the ECE can be considered according to the extent to which one wants to evaluate the model calibration. The class-wise ECE [125] computes a class-specific ECE for each class in a one vs. all setting:(22)$${\mathrm{ECE}}^{\mathrm{class}-\mathrm{wise}}=\frac{1}{C}\sum_{c=1}^{C}\sum_{m=1}^{M}\frac{\left|B_{mc}\right|}{N}\left|prev\left(B_{mc}\right)-conf\left(B_{mc}\right)\right|,$$
where *C* is the number of classes, $B_{mc}$ is the *m*-th bin for the *c* class, $prev(B_{mc})$ is the prevalence of the *c* class in the samples contained in the bin $B_{mc}$, and $conf(B_{mc})$ is the average confidence of the model in the *m*-th bin for the *c* class. Although the top-label ECE remains the most commonly used formulation of calibration error, others have been proposed in the literature to overcome its multiple shortcomings [126,127].

The Maximum Calibration Error (MCE) can be obtained by replacing the expectation over calibration errors by the maximum error across bins [127]. This metric may be adequate when accurate confidence estimation is critical. To evaluate the calibration of a regression model, we can execute the following procedure [116]:Run inference over dataset and record model predictions.For each threshold $p_{m}$:(a)Compute the probability ${\hat{p}}_{m}$ of a sample having a predicted Cumulative Distribution Function (CDF) value lower than $p_{m}$;(b)Compute the calibration error for the threshold.

The ECE of a regression model can be defined as the average calibration error over every threshold:(23)$${\mathrm{ECE}}^{\mathrm{regression}}=\frac{1}{M}\sum_{m=1}^{M}\left|p_{m}-{\hat{p}}_{m}\right|,$$
where
(24)$${\hat{p}}_{m}=\frac{\left|\left\{i=1,\cdots ,N\;|\;F_{i}\left(y_{i}\right)\leq p_{m}\right\}\right|}{N},$$
where *M* is the number of evaluated thresholds, *N* is the number of samples, $y_{i}$ is the GT value, and $F_{i}\left(y_{i}\right)$ is the predicted CDF value for sample $y_{i}$. Figure 14 shows the relationship between predicted confidence levels and true confidence levels in calibrated and uncalibrated models.

#### 3.1.2. Scoring Rules

Measuring calibration is insufficient to evaluate a probabilistic model. For example, a model that returns the marginal distribution of the target variable is calibrated and yet is not useful as a predictive model. Models should not only produce calibrated predictions but also assign high probability to the observed values and low probability to incorrect values. This quality can be measured using scoring rules. A scoring rule is a function that measures the accuracy of a probabilistic prediction based on both the prediction and the observed value [128]. A scoring rule is called proper if the expected cost is minimised by the true distribution of the data, or strictly proper if this minimum is unique.

Many commonly used performance metrics, such as accuracy or IoU, evaluate only the top confidence value predicted by the network. Since scoring rules evaluate probabilistic predictions instead of point estimates, they should also be used to evaluate models that perform UQ and output a predicted distribution.

The Negative Log Likelihood (NLL) is a strictly proper scoring rule which corresponds to the negative logarithm of the likelihood of the GT values under the distribution predicted by the model. NLL can be used both on classifiers and probabilistic regressors. Models that are better at estimating uncertainty should be better calibrated and consequently have a lower NLL, defined as:(25)$$\mathrm{NLL}=-\frac{1}{N}\sum_{i=1}^{N}\log f_{\theta }\left(y_{i}|x_{i}\right),$$
where $f_{\theta }\left(Y=y_{i}|x_{i}\right)$ is the likelihood assigned to the GT value $y_{i}$ by the model $f_{\theta }$, given input $x_{i}$.

The Brier Score (BS) [129] is a strictly proper scoring rule, exclusive to classification problems, that measures the difference between the predicted confidence of the model and the GT value. For binary classification, the BS is defined as:(26)$$\begin{array}{cc} \mathrm{BS} & =\frac{1}{N}\sum_{i=1}^{N}{\left(f_{i}-o_{i}\right)}^{2},\\ \;f_{i} & =p_{\theta }\left({\hat{y}}_{i}=1\right),\\ \;o_{i} & =I\{y_{i}=1\}, \end{array}$$
where $f_{i}$ is the predicted confidence, $p_{\theta }$ is the model being evaluated and $o_{i}$ is the binary class indicator. Figure 15 shows both NLL and BS as a function of the predicted confidence for the positive class in positive and negative samples.

The BS can be extended to multi-class classification problems as:(27)$$\begin{array}{cc} \mathrm{BS} & =\frac{1}{N}\sum_{i=1}^{N}\sum_{c=1}^{C}{\left(f_{ic}-o_{ic}\right)}^{2},\\ \;f_{ic} & =p_{\theta }\left({\hat{y}}_{i}=c\right),\\ \;o_{ic} & =I(y_{i}=c), \end{array}$$
where $f_{ic}$ is the predicted confidence, $p_{\theta }$ is the model being evaluated and $o_{ic}$ is the class indicator.

The Beta Family of Proper Scoring Rules [130], denoted here as BF (Equation (Equation 28)), generalises the two previous metrics:(28)$$\begin{array}{cc} \mathrm{BF} & =\frac{1}{N}\sum_{i=1}^{N}\left[o_{i}\int_{0}^{fi}t\;\omega \left(t\right)\;dt\int_{f_{i}}^{1}\left(1-t\right)\omega \left(t\right)\;dt+\left(1-o_{i}\right)\right],\\ \;\omega (t\;|\;\alpha ,\;\beta ) & =t^{\alpha -1}{\left(1-t\right)}^{\beta -1},\;\alpha >-1,\;\beta >-1, \end{array}$$
where ω(t|α,β) is a weight function which assigns different importance to different confidence areas. The scoring rule BF can also be written as:(29)$$\mathrm{BF}=\frac{1}{N}\sum_{i=1}^{N}\left[o_{i}\left(B\left(1;\alpha ,\beta +1\right)-B\left(f_{i};\alpha ,\beta +1\right)\right)+\left(1-o_{i}\right)B\left(f_{i};\alpha +1,\beta \right)\right],$$
where B(x;α,β) is the unregularised incomplete beta function:(30)$$B\left(x;\alpha ,\beta \right)=\int_{0}^{x}t^{\alpha -1}{\left(1-t\right)}^{\beta -1}dt.$$

Figure 16 shows different functions in the beta family for different values of the parameters α and β. Setting α=β=0, we obtain the NLL. Setting α=β=1, we obtain the BS. By setting α≠β, we can define different costs for false negatives and false positives. When α,β→∞, the scoring rule becomes a step function.

The Continuous Ranked Probability Score (CRPS) generalises the BS for evaluating the probabilistic prediction of a continuous variable [131]:(31)$$\mathrm{CRPS}\left(F,y\right)=\int_{R}{\left[F\left(x\right)-I\left(x\geq y\right)\right]}^{2}dx,$$
where F(x) is the predicted CDF, and *y* is the observed GT value.

### 3.2. Methods

One scaling method proposed by Platt [112] involves using a sigmoid function to map the scores into probability estimates. Using a Support Vector Machine (SVM) as the classifier for the experiments, the scaling parameters *A* and *B* of the shape $\hat{P}\left(c|x\right)$ (Equation (Equation 32)) are found by minimising the NLL of the data:(32)$$\hat{P}\left(c|x\right)=\frac{1}{1+e^{As(x)+B}}.$$

Nevertheless, the method can easily employ another optimisation strategy. An overview of the Platt scaling method is depicted in Figure 17.

The results presented in this work show that the combination of SVM and sigmoid maintains the sharpness of the original SVM and reaches a probability fidelity similar to the one obtained with the regularised likelihood kernel method.

The works on Histogram Binning, by Zadrozny and Elkan [113], present a calibration solution for Naïve Bayes (NB) that is easily extendable for other classification models. For classifiers where scores for a given class can be ranked with consistent accuracy (such as NB), a histogram can be produced by sorting the training examples according to their scores and dividing them evenly across *N* bins. When a new example is provided during inference, the corrected probability will be the fraction of the training examples in the bin corresponding to the new example’s classification score for that class. It should be noted that, despite not being required for NB to separate the training set into training and binning sets, this setting should be taken into account for most classifiers, where a larger set should be used for training. This should be performed because training a classifier involves learning considerably more parameters than setting bin probabilities. Furthermore, the authors suggest the usage of a relatively small number of bins (N=10) which, consequently, increases the number of examples per bin, in order to reduce the variance of the binned probability estimates.

By applying score binning (Binned NB), the publication [113] presents clear metric improvement on the KDD’98 dataset [132], which is very unbalanced, and in which misclassification errors have different importance. On one side, Platt’s scaling with sigmoid fitting is reported to not provide accurate probability estimations for a generic classification model, for instance, failing to properly correct NB. On another, Histogram Binning has the downside of requiring to optimally choose the number of bins, which is unlikely to be achieved on small or highly unbalanced datasets.

To address these issues, an intermediate approach, isotonic regression, is proposed [114]. This non-parametric form of regression learns a non-decreasing function capable of describing the mapping from classification score to class probabilities, assuming a similar ranking consistency between example scores and respective probabilities. The resulting function consists of constant stepwise portions, the values of which are obtained by fitting the data to minimise the mean-squared error as shown in Figure 18.

When the non-decreasing property is violated for two consecutive example scores (called Pair-Adjacent Violators (PAV)), the points are replaced by one, while also explicitly forming a bin, using those two examples as bin boundaries. The bin’s empirical class membership probability becomes the average of those examples’ classification scores. This operation is iteratively performed, starting from the examples with the lowest score and ending on the highest, repeating the whole process, progressively averaging the PAV, until the isotonic property extends throughout the training example space.

These methods are designed for binary classification problems, and their direct extension for a multi-class setting requires some adaptation. Two options are presented in [114], based on the multi-class all-pair code matrix, where the matrix constitutes an all-against-one setting. One solution involves the minimisation of the least-squares, with non-negative constraints [133] and the other, named coupling, iteratively minimises the log-loss instead [134]. However, in similar cases, there is a third option, where the estimates $P(c_{i}|x)$, for each binary classifier *i*, are normalised so that their sum totals 1. The authors further conclude that it is not clear which option presents better results and that when the estimates become calibrated, it makes less difference which method is used.

In 2017, Guo et al. [115] made the case that modern DL models are more prone to miscalibration. They further introduced temperature scaling as a new method for model calibration and compared several calibration approaches. The first hypothesis was mostly tested using variations of ResNet and the CIFAR-100 dataset. Comparing ResNet with the older LeNet, their difference is clearly visible in Figure 19: the average confidence is farther from the accuracy in the ResNet results, and the distribution of confidence values is much more skewed towards greater values (see histograms at the top).

The reliability diagrams demonstrate that ResNet is overconfident in its predictions in all confidence bins, while LeNet is underconfident with smaller calibration errors (red gaps).

A closer investigation, illustrated in Figure 20, suggests that the trend towards deeper and wider neural networks, while improving the classification error and producing more accurate predictions, might be contributing for the miscalibration of DL models. The use of batch normalisation and a weaker weight decay also contribute to the same problem in the conducted experiments.

In Figure 21, after 250 epochs, it is possible to observe a disconnect between the evolution of the test error and the test NLL. In fact, overfitting to the NLL, which is the loss the network is trained against, seems to be beneficial to the accuracy. Since the test error keeps decreasing, the behaviour of the test NLL is best explained by the network becoming overconfident in most of its incorrect classifications. Thus, the authors hypothesise that the network learns better classification accuracy at the expense of well-modelled probabilities. The overfitting of the network is revealed in the probabilistic error rather than the classification error.

The same publication goes on to suggest the use of a new calibration approach: temperature scaling (TS). Given the network’s logit $z_{i}$ (i.e., the non-probabilistic output) for each class *k*, the calibrated probability of the predicted class shall be:(33)$${\hat{q}}_{i}=\max_{k}\sigma_{SM}{\left(z_{i}/T\right)}^{\left(k\right)},$$
where $\sigma_{SM}$ is the softmax function, and *T* is the temperature value that softens the softmax for T>1. *T* is set to 1 and frozen in the training phase, which is followed by a post hoc recalibration step using a calibration set (the validation set can be used), where only the temperature parameter is trained in order to maximise the entropy of the output distribution, subject to certain restrictions. This technique does not change the predicted class, maintaining the original accuracy. It is a simpler extension of Platt scaling, as it only applies the multiplicative scalar to the logit as depicted in Figure 22.

After comparing temperature scaling with several calibration approaches (Figure 23), the results are surprisingly favourable for this simpler method, suggesting that network miscalibration is inherently low dimensional. Further tests are performed on standard vision and Natural Language Processing (NLP) datasets with different models: temperature scaling outperforms Histogram Binning, isotonic regression, Bayesian binning into quantiles, and matrix and vector scaling in most cases, in terms of ECE.

The idea that predictions with 70% confidence should be correct 70% of the time can also be transposed to the regression task. Kuleshov et al. [116] propose that a calibrated probabilistic regression model should output a distribution function such that a 70% confidence interval contains the GT 70% of the time. It is assumed that, given an input $x_{t}$, the forecaster *H* outputs a CDF $F_{t}$ targeting the label $y_{t}$. A Bayesian method is adopted in the publication, but other probabilistic methods can be applied to output a distribution function. A sufficient condition for a calibrated forecaster, which models the previously presented intuition, is defined as:(34)$$\lim_{T\to \infty }\;\frac{\sum_{t=1}^{T}I\left\{F_{t}\left(y_{t}\right)\leq p\right\}}{T}\to p,\;\forall p\in \left[0,1\right],$$
where *T* is the total number of samples in the dataset. The conditional statement $F_{t}\left(y_{t}\right)\leq p$ evaluates whether the GT value falls in the interval ]−∞,p], i.e., below the *p*-th quantile of the predicted distribution $F_{t}$. If this condition is verified, it follows that every other type of confidence interval will also be calibrated. A more compact form of the same condition is:(35)$$P(F_{X}\left(Y\right)\leq p)=p,\;\forall p\in \left[0,1\right].$$

Kuleshov et al. [116] then suggest the use of a recalibration auxiliary model R:[0,1]→[0,1] that transforms the CDF $F_{t}$ into a calibrated distribution $R\circ F_{t}$:(36)$$R\left(p\right)=P(F_{X}\left(Y\right)\leq p).$$

This definition holds that *R* should transform any predicted probability *p* into the empirically observed probability of a *p* confidence interval containing the label. For example, if only 80 out of 100 labels *Y* fall below the 95% quantile of the predicted distributions $F_{X}$, then the confidence value *p* of 95% should be calibrated to 80%. A recalibration dataset *D* should be constructed as:(37)$$D={\left\{F_{t}\left(y_{t}\right),\frac{\sum_{t=1}^{T}I\left\{F_{t}\left(y_{t}\right)\leq F_{t}\left(y_{t}\right)\right\}}{T}\right\}}_{t=1}^{T}.$$

The recalibrator *R* should then be trained on this dataset. The authors encourage the use of isotonic regression due to its monotonically increasing definition and non-parametric nature—it can learn the true distribution given enough independent and identically distributed (i.i.d.) data. The use of a separate calibration set for training *R* is advised to reduce overfitting. The improvement in calibration after using this method can be verified in Figure 24. The technique was proven to be more effective than concrete dropout and deep ensemble (previous methods), when applied to Bayesian linear regression or approximate Bayesian neural networks, for several UCI datasets [135].

The work of Wang et al. [117] tackles the open problem of calibration in Domain Adaptation (DA) of models learning higher classification accuracy by sacrificing well-calibrated probabilities in the process. To deal with this trade-off, the authors propose a method to attain more accurate calibration with lower bias and variance, in a unified hyperparameter-free optimisation framework, which they name Transferable Calibration (TransCal) in DA. They start by defining a calibration measure, Importance Weighted ECE, to be able to evaluate the calibration error in the target domain in a Transferable Calibration framework. They further propose a learnable meta parameter to keep reducing the estimation bias according to the theoretical analysis, and a serial control variate approach is produced to continuing to decrease the variance of the estimated calibration error. The applicability of this approach to recalibrate existing DA methods is reported as simple due to consisting of yet another general post hoc method.

Ding et al. [118] propose a variation on TS that is both image and location dependent, aimed at calibrating the multi-label semantic segmentation task. A smaller Convolutional Neural Network is used to define a different temperature scalar for each pixel/voxel. The network takes in both the input image and the logits output by the semantic segmentation model, which allows it to take into account the spatial correlations within each image. Similar to global TS, the calibration parameters (i.e., the auxiliary network’s parameters) are optimised with respect to the NLL on the validation set, in a post hoc step.

Experiments on four different datasets and tasks proved local TS to be superior to global TS, image/sample-dependent TS and other classic calibration methods. In general, the calibration error was significantly reduced both in the boundary areas between different classes, which are areas of increased uncertainty, and in random local patches of the image.

Tomani et al. [119] propose a parameterised TS approach, where the temperature is parameterised by an auxiliary neural network that considers the original network’s output logits in order to predict a fitting temperature. The post hoc calibrator for a trained neural network h(X) is fitted by optimising a squared error loss $L_{\theta }$ with respect to the weights of the neural network (*g*) parameterising the scalar temperature θ as:(38)$$L_{\theta }=\frac{1}{N}\sum_{n=1}^{N}\sum_{c=1}^{C}{\left(I_{nc}-\sigma_{SM}{\left(z_{i}/g_{\theta }\left(z_{i}^{s}\right)\right)}^{\left(c\right)}\right)}^{2}.$$

Here, $I_{nc}$ is 1 if sample *n* has true class *c*, and 0 otherwise; $\sigma_{SM}$ is the softmax function; and $z_{i}^{s}$ are the unnormalised sorted logits. This allows the recalibration network to decrease the prediction confidence in incorrectly predicted samples and to increase the prediction confidence in correctly predicted samples, thus reducing the calibration further than is possible with a global temperature parameter.

Joy et al. [120] propose a different approach for dealing with the issue of individual samples having fairly different contributions in regards of the calibration of model confidences, specially taking into account the model’s ability to correctly predict them. Calibration methods, such as temperature scaling [115], tend to reduce the individual confidences of the predictions without considering the correctness of classification for a given input. To tackle this effect, individual temperature values *T* can be predicted on a per data-point basis, permitting T>1 for correctly classified samples, and T<1 for the incorrect ones, effectively increasing or decreasing their confidence on an individual need and with varying amounts.

Here, the term prediction stems from the fact that the overall architecture includes a temperature prediction module, which consists of a small neural network, in this case a learnable VAE encoder q(z|ϕ(x)), plus a Multi-layered Perceptron that outputs *T*.

This temperature predictor extracts information from the main network, in the form of features, which should learn how to obtain the required information for class prediction while also containing a notion of the associated confidence. This setup seems to have been providing promising results for a variety of tasks that require model calibration [136]. The authors argue that unlike a standard auto-encoder, this strategy creates a mechanism for obtaining a likelihood on the latent codes. These latent likelihoods could then be used to predict the temperature value.

It is also reported in this work that, by introducing a prior for each class (parameterised by ${\lambda_{k}}^{K}$), issues with clustering individual classes in lower likelihood regions of the latent space are prevented, thus allowing the classes to cluster individually.

### 3.3. Applications in Autonomous Driving

A study on the calibration of a probabilistic LiDAR object detector concludes that the naïve use of direct modelling for the UQ of the bounding box regression task generates miscalibrated uncertainties [121]. A modified version of PIXOR, a one-stage LiDAR object detection network, is used in the experiments, with BEV maps as input. Following the direct modelling approach, the experiments are conducted by adding an extra output vector and modifying the loss in order to model the vector as the variance (or uncertainty) of each parameter. This assumes a multi-variate Gaussian distribution of predictions. Therefore, the overall regression uncertainty is computed as a diagonal covariance matrix, with the simplified assumption of independently distributed bounding box parameters.

Isotonic regression and temperature scaling are employed in order to improve the calibration of the regression task. A recalibration set is built from the validation set for this purpose. The optimal temperature for each regression parameter is found by minimising the NLL on this set. Furthermore, an addition to the training loss is proposed to produce better calibrated variances, incentivising the predicted variances to match the regression errors. This calibration loss term can be added to the normal training of the network, without the need of a further recalibration step. Isotonic regression changes the assumed probability distribution family (Gaussian in this case), while the other methods keep it the same.

This work finds that all the methods are effective in reducing the ECE of the model, using both KITTI [71] and nuScenes [109] datasets, similar to the improvement observed in Figure 25. The use of the proposed calibration loss term is also found to improve the detection performance.

Brickwedde et al. [122] presents another approach, where a smaller DL model is made to correct confidence inconsistencies across a main model’s predictions. In this case, DepthNet predicts pixel-wise depth distributions as a mixture of Gaussians (mean, variance, and weight of each component), and CalibNet is trained to take the variances and weights of each component and recalibrate them. DepthNet and CalibNet are both trained on KITTI raw sequences [99] and evaluated on the KITTI optical flow dataset [137], Make3D [59], and Cityscapes [103].

Wang et al. [123] propose an extension of selective classification to model calibration, which is named Selective Scaling. The method introduces a binary classifier as a selector to categorise predictions into correct/incorrect for scaling as depicted in Equation (Equation 39). It then focuses on smoothing misprediction logits to alleviate the miscalibration from overconfidence:(39)$$\hat{p}=\left\{\begin{array}{cc} \sigma_{SM}\left(z/T_{1}\right), & \mathrm{if}\;\hat{y}\neq y\\ \sigma_{SM}\left(z/T_{2}\right), & \mathrm{otherwise}. \end{array}\right.\;$$

Here, $\hat{p}$ corresponds to the calibrated probability vector, $\sigma_{SM}$ corresponds to the softmax activation function, *z* is the logit vector before activation, $T_{1}$ is a temperature to smooth the logit distribution, and $T_{2}$ is a temperature to sharpen the logit distribution, so $T_{1}>T_{2}$.

Furthermore, the authors propose an extension of the ECE that focuses on semantic segmentation. For this version, the ECE is calculated over each image *I* first, and then the values are averaged across the *N* inference images:(40)$$\mathrm{ECE}=\frac{1}{N}\sum_{I=1}^{N}{\mathrm{ECE}}_{I}.\;$$

Several in-domain and domain-shift tests are made on the semantic segmentation task, using multiple datasets across various applications. The AD datasets used are BDD100K [107], Citycapes [103] and SYNTHIA [138].

The work also presents several experiments where the effects on calibration and accuracy are studied upon the variation of model depth and width, input crop size, testing with multi-scale input and domain shift calibration.

## 4. Uncertainty Applications

One of the greatest challenges of deploying autonomous vehicles in the open-world context is that there are many events with a low probability of occurrence, due to the long-tail nature of the domain, and there are a priori unknown data that are not encountered during training. Neural networks, like other machine learning models, struggle with making reliable predictions, i.e., proper generalisation, for samples which lie outside the training distribution. In particular, training samples that are included in the training dataset are considered ID, whereas samples from distributions different from the training distribution are considered out-of-distribution (OOD). Under-represented samples in the training set, lying on the tail of the class distribution, can also be considered OOD.

The differences between ID and OOD samples may be due to one or more various factors, such as different sensors or sensor configurations, different weather or visibility conditions, different geographical locations, different appearance of objects of interest or even new classes of objects, not present during training. Furthermore, it is not possible to comprehensively specify all the ways in which a sample may differ from the training set, as some of them may not even be known a priori. Regardless of the source of their oddness, their unavailability during training should increase the uncertainty of its predictions. Therefore, it is important that perception models have some way to signal that a given input is sufficiently different from the training dataset such that its predictions should be treated with additional caution. These inputs can then be handled differently depending on where the method is applied. During the development phase, inputs signalled as OOD might be manually labelled and added to the training dataset, in an active learning framework. During operation, the predictions of the model for these inputs should not be trusted fully and, therefore, a fail-safe mechanism of OOD detection should be engaged.

### 4.1. Out-of-Distribution Detection

When object types that were not available during training are presented in a real-world setting, the model should not assume that it belongs to a seen class. Similarly, in a case of a covariate shift, i.e., if conditions are substantially different from those that the model trained on, and it is not able to be reliable, the model should signal its inability to perform adequately. OOD situations like these must be reported by the models themselves in order to provide the dependable predictors necessary for safe Autonomous Driving.

OOD detection has not been extensively explored for AD applications. Therefore, in this section, we first describe the works developed on the general OOD application, which are mostly applied to image classification using some of the most popular public datasets, such as MNIST and CIFAR. After that, we report more recent works where the fundamental concepts are adapted and demonstrated in the field of AD, and more specifically to object detection and LiDAR data. Methods for OOD detection can be divided into five categories:**Supervised classification** methods for OOD detection involve training a classifier to distinguish ID samples from OOD samples. Alternatively, an existing classifier can be modified by adding an additional OOD class. This is a simple approach which has the disadvantage of requiring annotated OOD samples to be present in the training set, which is not always a viable solution.**Distance-based** methods detect OOD samples by their distance to the set of ID samples [17,52]. In general, distance in a semantically meaningful space should be higher between OOD and ID samples than between two ID samples.**Self-supervised** methods use some self-supervised task to detect OOD examples. Self-supervised methods typically show lower performance in samples that differ from the training set, allowing ID and OOD. One example of a self-supervised task is reconstruction.**Generative models** aim to model the distribution of the data themselves. If they allow for computing a likelihood of the data, ID data should present higher values of likelihood or pseudo-likelihood than OOD data [139].**Uncertainty-based** methods use some UQ method to separate ID from OOD data [140].**Energy-based** methods attempt to assign higher energy scores to samples with a lower likelihood of occurrence, allowing the separation of OOD and ID samples.

However, the most recent methodologies can combine more than one of these categories. Table 3 summarises the OOD methods that are described in this section.

#### 4.1.1. Methods

Early work concerning OOD started to show the impact of the base usage of the maximum softmax score to distinguish between ID and OOD samples [141]. A proposed improvement to this baseline included an abnormality classifier, in which a decoder reconstructed the input and only a downstream abnormality sub-network was trained on both ID and OOD data. The extent of training scenarios ranged from image with seen and unseen classes and styles (MNIST, CIFAR-10 and CIFAR-100, vs. Scene Understanding/SUN [162], notMNIST, binarised, added noise and transformed versions), NLP tasks (sentiment classification, text categorisation, part-of-speech tagging) and even Speech Recognition (TIMIT vs. added noise versions).

Several works have explored the ability to detect OOD samples (in the unseen class scenario), making use of the dataset of hand-drawn numerals, MNIST, against the alphabetical notMNIST. That is the case of Lakshminarayanan et al. [21], who also applied their experiments to the dataset of Street View House Numbers (SVHN) [163] and considered the classes of the label diverse CIFAR10 as out of distribution. In an effort to demonstrate the scalability of the methods, this work also separated categories from ImageNet [164], treating dog classes as known and non-dog examples as unknown classes. Among other findings, the authors concluded that models trained with adversarial examples and in an ensemble setting outperformed the same base model employing MC Dropout, in terms of both accuracy and accurate evaluation of uncertainty.

A later work compares multiple literature options concerning uncertainty evaluation in the i.i.d. setting and under distributional shift [68]. Compared approaches include Maximum Softmax probability [141], temperature scaling [115], MC Dropout [25], ensembles [21], Stochastic Variational Bayesian Inference [7], and approximate Bayesian Inference for the parameters of the last layer only, separately evaluating the effect with mean field stochastic variational inference or with Dropout [165]. The work analyses how the same model, separately employing each of these schemes, is affected when subjected to test data that are either a corrupted, perturbed version of the training data (shifted version) or constitutes unseen labels. The work uses the, by now, popular OOD dataset pairing of MNIST vs. notMNIST, and analyses the shift effect over CIFAR-10 and ImageNet data, as well as over the 20NewsGroups [166] and the Criteo Display Advertising Challenge [167] datasets. To evaluate the impact of increasingly shifted data in uncertainty, the metrics NLL, BS and ECE are used.

The authors report several findings concerning OOD, among which include that better calibration and accuracy on the i.i.d. test dataset does not often correspond to better calibration under dataset shift and that last layer Dropout performs better than MC Dropout. Furthermore, they claim that approaches using the epistemic uncertainty tend to outperform calibration (on i.i.d. validation) with temperature scaling when larger dataset shift occurs, and that deep ensembles perform best among the approaches tested.

Another work [142] brings to light a method to fix the limitations of the behaviour of ReLU networks since, as demonstrated by [168], these classifiers generate progressively more confident predictions the further they move away from the training data as depicted in Figure 26. To do this, they suggest the use of a Certified Certain Uncertainty (CCU), where GMMs are used to controllably produce data far away from the training distribution and ensure that the used classifier reports uniform confidence for an OOD sample. The experiments performed employ MNIST, FashionMNIST [169], SVHN, CIFAR-10 and CIFAR-100 datasets, among several others and the addition of noise.

OOD detection closely relates to the Open-set Recognition (OSR) problem, where there is an incomplete knowledge of the world at training time and new classes are present at testing time. This dichotomy has been tackled by analysing different ways to estimate predictive uncertainty, namely, obtaining epistemic uncertainty, and contrasting those findings with Extreme Value Theory (EVT) results [143]. In particular, the authors of the work empirically find that the EVT approach, leveraging sample rejection over single class fitted Weibull distributions, outperforms OOD detection with predictive uncertainty, for several classification tasks. Additionally, they employ a generative architecture for approximating both the label distribution p(y) and the data distribution p(x), stating it might still be required for open set recognition in classification, despite previous research showing the limitations of the generative approaches that only model the data distribution in distinguishing between seen and unseen data [170]. These open set experiments are conducted by training with the FashionMNIST dataset and using as unseen data the MNIST, K-MNIST, CIFAR-10, CIFAR-100, SVHN datasets and the spectrogram processing of the AudioMNIST [171] dataset.

The usage of softmax functions in discriminative models has been shown to significantly increase overconfidence of predictions under covariate shift and on OOD data [144]. This is presented by providing an analysis of different failure modes of softmax cross-entropy. Additionally, the authors show that logits downstream from linearly transformed embeddings are guilty of increasing output miscalibration. To mitigate this effect and improve performance, the authors suggest employing a one-vs-all formulation (OVA), effectively leveraging *K* binary classifiers, where *K* is equal to the number of trainable classes, each parameterised by a sigmoid activation. They further report that performance is enhanced when a distance-based logit representation (DM) is added and used to encode uncertainty as a measure of distance to the training manifold as shown in Figure 27. In this case, the probability distribution becomes for neural network embeddings $f_{\theta }\left(x\right)$:(41)$$\mathrm{POVA}-\mathrm{DM}\left({\hat{y}}^{\left(k\right)}|x\right)=\frac{2}{1+exp(-∥f_{\theta }\left(x\right)-w_{k}∥)}$$

However, they warn that the distance-based one-vs-all formulation does not scale well, providing comparatively lower results than softmax cross-entropy on tasks with a larger number of in-distribution classes (ImageNet), most likely due to the inability to provide balanced training batches.

Mohseni et al. [145] improve OOD detection by performing optimised representation learning. Instead of being manually designed, the authors propose a method to select the most effective pretext tasks and shifting transformations. The authors demonstrate that the optimal set of transformations depends on the ID training data distribution, and should therefore be learned for each dataset, without the need of any OOD training samples. The set of possible shifting transformations include geometric (translation and rotation) and non-geometric ones (blurring, sharpening, colour jittering, Gaussian noise, and cutout). The complete model has a common feature extraction backbone and multiple 2-layer fully connected heads, one for the main classification task and one for each auxiliary task, which includes the main classification after each transformation, and a self-supervised classification of the transformation type. The weighted sum of the output of all supervised tasks (i.e., the main classification task and the auxiliary ones) can be interpreted as an ensemble score. The OOD detection score is the KL divergence between the ensemble score and the uniform distribution. The proposed method outperforms several baseline and SOTA OOD detection methods on multiple combinations of ID and OOD datasets. Additionally, the authors develop a method that predicts the transformation that is applied to the training data. This method, in the self-supervised or fully supervised version, improves both ID classification and OOD detection.

The work of Djurisic et al. [146] is centered on the premise that the representations learned for ID samples by the model are more robust than those of OOD samples. As such, the authors propose detecting OOD samples by purposefully degrading object representations and comparing the impact of the network’s new output with the original one. A similar concept is used by Hebbalaguppe et al. [147], although instead of directly applying distortions on the network’s latent space, they apply it in the original space. While in the previous paper the shaping is applied only at inference time, in this one, the OOD detector is trained with a (K + 1) classifier, where K is the ID classes and the additional class corresponds to the proxy OOD data.

Liu et al. [148] propose the use of an energy score based on the logit outputs of the network to differentiate between ID and OOD samples, in an image classification setting as illustrated by Figure 28. The energy score is defined as follows:(42)$$E\left(x;f\right)=-T\cdot \log\sum_{i}^{C}e^{f_{i}\left(x\right)/T},$$
where *f* represents the classification model, $f_{i}\left(x\right)$ is the logit value of the *i*-th class for input image x, and *T* is the temperature. The authors demonstrate using multiple OOD datasets such that the energy score leads to better OOD detection compared to using the baseline softmax score of the predicted class, as well as other more complex SOTA methods. These results can be achieved with any pre-trained classification model. However, a specific training policy can be employed by adding an additional term to the cross-entropy loss, incentivising the model to output higher values of energy when OOD samples are fed into it. The parameter-free nature of this approach renders it a simple but effective solution.

Huang and Li [149] show that the performance of OOD detection techniques deteriorates as the number of classes increases. As such, they propose a grouping strategy during training and inference that decomposes a high-dimensional class space into groups of related classes. The group mapping can be defined manually, randomly or by applying k-means over the features from the network’s last layer. Group softmax is used to compute the class probabilities within each group. All groups include an *others* class, which represents any class outside of the respective group. The proposed OOD detection score, Minimum Others Score (MOS), leverages the predicted probabilities of the *others* classes, and surpassed multiple baselines on different OOD datasets.

Sun et al. [150] have observed that the internal activations of neural networks display highly distinctive signature patterns for OOD distributions. Regarding the distribution of the activations of the penultimate layer of the ResNet50 trained on ImageNet, the mean activation for ID data has a near-constant mean and standard deviation, while OOD data have significantly larger variations across units with sharp positive values. This phenomenon is also found in other OOD datasets. Therefore, the authors propose a simple and effective technique called Rectified Activations (also known as ReAct), in which the outsized activation of a few selected hidden units can be attenuated by truncating the activations at an upper limit c>0. The use of ReAct improves the separation of ID and OOD and generalises effectively to different network architectures and different OOD detection scores.

As previously mentioned, concerning several works, MC Dropout has been popularly used for OOD detection, as training with it effectively implements variational inference and, keeping the Dropout active during test time, enables sampling the posterior distribution, like a pseudo-ensemble. However, works have shown the contributions of this method, measuring uncertainty only at the final outputs, or even only using Dropout at test time on the final layer, leaving potentially useful intermediate information untapped [151]. The logic of the approach concerns an expected reduction in activations in intermediate layers due to logits/softmax inputs of OOD samples presenting lower norms [141].

Hence, Nguyen et al. [151] measure the dispersion from randomised embedding features, with the assistance of the Cosine Distance, capturing not only their norm but also the angular information of the embeddings, removing confounding effects caused by systematic norm differences. The authors find better separation between in- and out-of-distribution samples and wider spread concerning OOD. They further claim that, despite helping, additional regularisation methods are not needed, and due to the way this approach is achieved, virtually no overhead cost is required in terms of memory or compute power. During their experiments, the LeNet5 models are trained in MNIST, with OOD datasets such as K-MNIST, notMNSIT, FashionMNIST for image classification. For a language classification task, a Char-CNN uses the WiLI dataset corpus, using some language sets for training and others as OOD. Finally, a Malware detection task is also performed, employing a Bayesian MalConv model trained on the EMBER2018 [172] dataset of binary executable files and, for OOD, a subset from 2012 and a dataset of malware that targeted a Brazilian financial entity.

Something in common to most works concerning OOD and the general study machine learning is that well-known samples tend to cluster together rather tightly, in high-density regions in the latent space. On the other hand, unknown samples or new patterns generally present a scattered behaviour, being spread across low-density regions. Naturally, research has been conducted towards figuring out how to leverage this notion to apply to the Open-set Object Detection (OSOD) problem, in particular, without resorting to complex post-processing.

In the work of Han et al. [152], the Open-set Detector (OpenDet) expands low-density regions by producing more compact features for known classes. A modified Faster R-CNN is used as the backbone for a Contrastive Feature Learner (CFL), and an Unknown Probability Learner module (UPL). The CFL produces more compact intra-class features, while increasing the inter-class separation. The UPL optimises the boundaries separating each known class and an extra class aggregating unknown objects. The PASCAL VOC [92] and the MS COCO [93] datasets are used in different split configurations to build an OSOD benchmark. The developed method is compared to approaches such as Faster R-CNN [173], Dropout Sampling [174], ORE [175], and PROSER [176], with the assistance of the following metrics: Wilderness Impact [177], corresponding to the percentage of unknown objects misclassified as known classes; Absolute Open-Set Error (AOSE) [174], the number of misclassified unknown objects; and the Mean Average Precision (mAP) for known and unknown classes. The results show that OpenDet outperforms other methods by a large margin.

OSR is a similar task to OOD detection that aims to identify whether or not a test sample belongs to one of the semantic classes in a classifier’s training. A recent work [153] finds that the performance of the OSR is highly correlated with the model’s accuracy in the closed set. The correlation is also held across several loss objectives and architectures as shown in Figure 29. This is surprising because stronger closed-set classifiers are more likely to overfit to the training classes and perform poorly in OSR. Another relevant contribution of this paper is the introduction of a ‘Semantic Shift Benchmark’, which consists of two ‘Easy’ and ‘Hard’ test sets based on the semantic similarity of the open-set categories to the training classes. This setup is more appropriate to capture the model’s performance to detect semantic novelty in contrast to other forms of distribution shift.

Yang et al. [154] introduce the task of full-spectrum OOD detection by using four types of data: training ID, covariate-shifted ID, near-OOD, and far-OOD. To improve the detection performance, the authors develop a feature-based semantics score function—SEM—composed by one measure based on high-level features containing both semantic and non-semantic information and another on low-level feature statistics only capturing non-semantic image styles. The non-semantic part is cancelled out to leave only the semantic information. Another important contribution of this paper is the creation of three full-spectrum OOD open-sourced benchmark datasets.

Blei et al. [155] present an OOD detecting approach for 2D object detection, that proposes the inclusion of the margin entropy (ME) loss, which is, by definition, high when the model estimates high confidence for OOD samples, and does not affect the training of ID samples. Furthermore, they introduce separability as a metric for detecting OOD samples in object detection. The authors report an increase in OOD detection, over using standard confidence scores, while keeping the same inference time.

Open Long-Tailed Recognition++ (OLTR++) [156] focuses on increasing the robustness of long-tail cases’ detection by relating embeddings of common and rare cases. The sensitivity to the open class is achieved through a dynamic calibration of the embedding related to the visual memory. The proposed algorithm distinguishes between rare cases (tail) and ID and OOD (open).

#### 4.1.2. Applications in Autonomous Driving

The direct application of OOD detection for automotive scenarios has been explored by more recent works. Nitsch et al. [157] leverage an auxiliary Generative Adversarial Network (GAN) to produce OOD samples during train time, and the classifier receives both ID and OOD objects, assigning low confidence to samples below the decision threshold. Next, Gaussian distributions are obtained for each ID class, which are used as reference distributions against other objects’ distributions, during inference time, ultimately identifying OOD objects based on that distance. The authors experiment with several post hoc confidence measures, namely, the largest softmax probability amongst all classes, mutual information from MC dropout sampling, cumulative density function of the $\chi^{2}$ distribution, and cosine similarity, having obtained the best results with the latter. The datasets used are KITTI and nuScenes. Feng et al. [158] also use a generative technique: a VAE, which models the data distribution as Gaussian features (with mean μ and variance $\sigma^{2}$). The authors apply the VAE to a 3D optical flow space and compute the KL divergence in the horizontal and vertical subspaces of the optical flow between the encoder output and the prior to obtain an OOD score, which allows OOD detection. The approach is tested in the nuScenes dataset with Autodesk Maya software’s dynamic effects and Synthia [138], a synthetic dataset of driving scenarios for semantic segmentation.

Hosoya et al. [159] propose a more feasible definition of the OSOD task, which aims only at the detection of unknown objects belonging to the same super-classes as known classes (e.g., Vehicle and Traffic Signal are examples of super-classes). This simplification solves the issue of determining what actually constitutes individual objects, allowing the computation of average precision for known and unknown classes. Moreover, objects from the same super-class will have more visual similarity, making the detection of unknown objects easier. This approach is applied to SOTA methods [152,175,178] and to a simple baseline based on prediction uncertainty. The results show that SOTA methods attain only limited performance, similar to the baseline. Several datasets are used, including the Open Images Dataset v6 [179], COCO [93], and Mapillary Traffic Sign Dataset [180].

The detection of OOD data in real time has been demonstrated by [160], using a mobile autonomous robot that navigates a miniature town and stops in the presence of unknown, novel samples (images with simulated snowfall in this case). The detection is made by a YOLO object detection network in images, and the OOD detector is based on the optical flow proposed in [158], that uses a VAE to identify environmental motion not present in the training set.

The work of Huang et al. [161] is the first to tackle OOD detection in 3D point clouds, making use of PointPillars [181] networks enhanced by several approaches, namely, Max-softmax, Uncertainty estimation, Mahalanobis distance, OC-SVM, and NF. The authors propose a definition of OOD for object detection, enumerating all possible cases regarding foreground (ID and OOD objects) and background detection and classification. The evaluation is performed over the KITTI dataset, complemented with additional real and synthetic OOD objects. Their main conclusions revolve around the notion of individual method performance for OOD detection heavily depending on the type of objects in question, which leads to highly variable results. As a consequence, they suggest that a more robust OOD detector should require the combination of multiple approaches.

### 4.2. Active Learning

Active learning (AL) automatically signals informative samples to be used in training. These samples can include OOD cases or insufficiently represented objects. This precocious identification helps to focus annotation efforts and thus reduce costs and provide a more balanced dataset. In the automotive field, the scale of captured data with a fleet of cars, the cost of annotation and the long tail of events of the self-driving problem motivate the conception of efficient and automated ways of selecting data for annotation. The most informative samples should be picked: a diverse set of scenarios, including rare and difficult situations that might boost the model’s performance. Manual curation by experts can certainly address the diversity of the dataset but does not necessarily find the data that the model has the most trouble with, as well as not being a scalable proposition.

Therefore, AL can solve this problem, as it iteratively increments the labelled dataset with sets of samples that provide the most relevant information for the training loop, hopefully improving the predictive performance with the least amount of annotation effort. Being a more recent topic than OOD detection, there are more works that apply AL to object detection and also to the LiDAR data of driving scenes. Table 4 summarises the AL methods that are described below.

#### 4.2.1. Methods

In AL, it is essential to prevent class imbalance issues such as when mostly only the sample dominant classes are selected. Zhang et al. [182] propose a method for balancing the selected classes within each batch during the AL process. The authors sort the observations according to uncertainty scores and apply a bisection procedure to find consecutive pairs of points with differing labels. However, this method requires sequential and synchronous annotation for each batch, which limits potential parallelisation efforts from multiple annotators.

In balanced active learning [183], the authors propose to reduce the effect of class imbalance in the AL process by identifying object similarity through the values of the embedded space of a self-supervision optimised sub-module.

Yoo and Kweon [184] follow a different approach: a sub-module is added to the model such that the sample’s loss is predicted. This sub-module can then be used to guide the AL process by selecting the samples with larger predicted loss. On the other hand, Geifman and El-Yaniv [185] suggest guiding this process by identifying outliers, leveraging several activation outputs throughout the model, instead of just the output model uncertainty.

For the task of pose estimation, the authors of VL4Pose [186] suggest using the model output inconsistencies to detect OOD samples, which are subsequently annotated. This concept of using model inconsistencies resembles somewhat the logic behind calculating and leveraging epistemic uncertainty for AL purposes.

Li and Alstrøm [187] elaborate on the importance of uncertainty calibration for efficient and effective AL. For that, several calibration quantification metrics are applied with different acquisition strategies for an image segmentation framework. The authors conclude that annotating only regions, instead of full images, improves model calibration, which boosts the effectiveness of the AL framework.

Although the majority of AL methods focus on entire frames, Lin et al. [188] propose the calculation of uncertainty metrics for the different regions of the point cloud. Thus, this approach is only suited for architectures that work directly over the point cloud representation. In particular, the authors employ PointNet++ as their experimental baseline and test the methods in two subsets of the AHN3 dataset [193], which contains point clouds captured by airborne laser scanning of Dutch cities. An incremental fine-tuning strategy is implemented by using both old and newly selected tiles and by using the parameters from the last AL iteration as the initialisation of the current model, in order to maintain the knowledge from previous training efforts.

#### 4.2.2. Applications in Autonomous Driving

The use of UQ for AL in the object detection task is researched in Haussmann et al. [189], with images for AD from an internal large scale research dataset. The authors suggest the use of an ensemble and the quantification of uncertainty in order to score unlabelled images and select a new set for annotation. A new supervised training iteration is then performed with the incremented dataset as Figure 30 illustrates. Using an ensemble of eight single-stage object detectors with a U-Net backbone, they show that multiple measures of uncertainty, including entropy (linked to total predictive uncertainty) and mutual information (linked to epistemic uncertainty), can successfully be used to select samples to label, improving model performance. The detection of objects see an improvement of 3× for pedestrians and 4.4× for bicycles in night-time using AL relative to a manual selection of data.

AL for 3D LiDAR-based object detection is implemented and tested in Feng et al. [82], also using predictive UQ to score the data informativeness of object proposals. Using the KITTI dataset, proposals are generated using a 2D object detector on the images, due to the simpler nature of the 2D task. A small section of the point cloud, containing each object, is projected onto the front-view camera image creating depth and intensity maps, through a 3D frustum. These are input to a convolutional network, which in turn outputs the objectness score and object location parameters. The network is trained from scratch in each iteration, although an iterative fine-tuning approach should be more scalable and practical. MC Dropout and deep ensembles are both employed to obtain multiple inferences per proposal. Shannon entropy and mutual information are then used as query functions, measuring the predictive uncertainty and the epistemic uncertainty, respectively, of the classification scores. The use of these methods is shown to provide a more useful ranking strategy than the single uncalibrated softmax output, while also allowing a reduction in the labelling effort for the classification task by up to 60%, compared to a random sampling baseline.

In Jiang et al. [190], the authors use rare example mining techniques in order to identify rare objects for annotation purposes. This is performed by estimating latent feature density estimation with a Normalising Flows model. The work uses the Waymo Open Dataset [110].

While performing AL on sequential data, such as video, it is crucial to take into account the need to selectively annotate frames from different sequences to mitigate the risk of producing highly biased and overfitted models [191]. By actively avoiding the simultaneous selection of very similar frames, researchers can ensure a more diverse and representative training dataset, enhancing the generalisation capabilities of the resulting models. In *One Class One Click* (OCOC) [192], the point cloud annotators’ work is reduced, as the model identifies sub-clouds and only requires annotations over these few point clusters.

## 5. Conclusions

In this document, we have summarised the most relevant works concerning reliability of Deep Learning models, with a focus on AD tasks.

Uncertainty Quantification is a challenging and essential problem that needs to be tackled in order to enable reliable and safe Autonomous Driving. In a relatively short amount of time, a wide variety of different techniques have surfaced to mitigate this problem, each with its own benefits and drawbacks. One problem we identified in the literature is that some papers use seemingly incompatible definitions of uncertainty, considering it model-independent in some cases and model-dependent in others, without clearly specifying which definition is being used. This can compromise the interpretability of the methods and, therefore, we recommend future researchers to include this clarification. A relatively underdeveloped aspect of many UQ techniques in the literature is their ability to run in real time. This property is especially important in the field of AD, where self-driving agents must make safety-critical, near-instantaneous decisions based on limited information, and so must quickly and accurately assess the uncertainty inherent to their perception of the environment.

Several metrics allow the evaluation of model calibration in both classification and regression problems, such as semantic segmentation and object detection. The accuracy of probabilistic predictions can be evaluated with scoring rules, while calibration can be directly measured by different formulations of calibration error. Sharpness, on the other hand, evaluates the informativeness of the prediction. With these metrics, it becomes possible to evaluate the reliability as well as the correctness of perception models, which is key to ensure the safety of AD vehicles. The improvement in model calibration turns the output confidences or variances into more trustworthy values with statistical relevance, increasing the reliability of the perception model. Most methods run in a post hoc step using a hold-out calibration set, although a calibration term can be added to the main training loss. Despite the lack of many works on calibration applied to AD perception in the literature, several SOTA methods, namely temperature scaling and its variants, look promising due to their simplicity, low computational effort, and lack of impact in the highest confidence class/value.

The topic of OOD detection has started to gain traction in recent years. Regarding uncertainty-based methods, MC Dropout and its variations, as well as ensembles and calibration approaches, are predominant in the literature. With increasing frequency and depth, researchers have built their network architectures with uncertainty-inspired concerns and concepts. Nevertheless, more recent concepts and revamping of more classical ones have also been the subject of interest. The conversion of multi-class to multiple binary classification in a one-vs-all formulation has been considered to be of importance, as well as making a new sub-distribution based on the closed-set outliers (EVT). Distance-based clustering has enabled to reduce membership probabilities far away from training data and to avoid the threshold-based overconfidence issue. The usage of cosine distances has allowed for the representation of embedding information and the reduction in angular-induced confounding effects. Several works propose new OOD scores, and others explore generative techniques to improve OOD detection. In the object detection task, the literature reflects the complexity of the problem and the use of different definitions, evidencing the need for a consensus in the scientific community. Despite these developments and a few recent works in the field, the outlook for OOD detection still leaves untapped research potential regarding AD, particularly where it concerns LiDAR-based solutions.

Active learning has shown to be a very important topic in DL and especially in AD, due to performance dependence on the quantity and diversity of labelled data. AL has been applied not only to the classification of images but also to object detection and semantic segmentation with LiDAR data of automotive scenes. Uncertainty is the most used indicator of the unlabelled samples that are most needed to be annotated. Other indicators are based on the loss, likelihood, or values of the embedded or latent space. Some methods have been developed to tackle class imbalance in AL, including graph-based and self-supervised strategies. Finally, annotation by region rather than by full image or point cloud and also uncertainty calibration has been proven to contribute to the efficiency and effectiveness of AL.

To conclude, we hope this survey can guide further research on safe AI algorithms in Autonomous Driving.

## Figures and Tables

**Figure 1 entropy-26-00634-f001:**
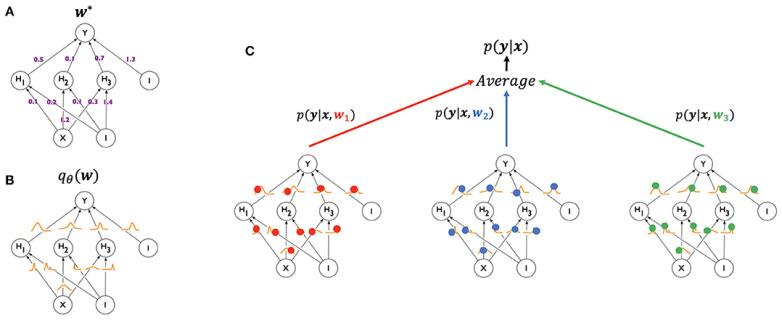
Illustration of model inference with a Bayesian neural network (from [6], image by McClure et al. [6], licensed under CC BY 4.0 https://creativecommons.org/licenses/by/4.0/, accessed on 13 June 2024. Modified from Blundell et al. [7]). A standard neural network (**A**) contains a point estimate, or single value ($w^{*}$) for its weights. A Bayesian neural network (**B**), on the other hand, maintains a distribution over its weights, $q_{\theta }\left(w\right)$, parameterised by θ. In the inference phase (**C**), this weight distribution can be sampled from, resulting in different points in weight space ($w_{1}$—red, $w_{2}$—blue, $w_{3}$—green). These points can then be treated as an ensemble by averaging their predictions.

**Figure 2 entropy-26-00634-f002:**
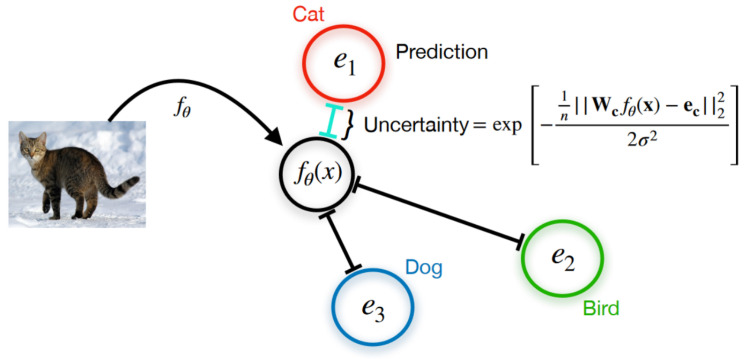
Illustration of the DUQ methodology (from [17]): the feature space representation of an input is assigned to the closest class centroid. Uncertainty is measured through the distance to the assigned centroid.

**Figure 3 entropy-26-00634-f003:**
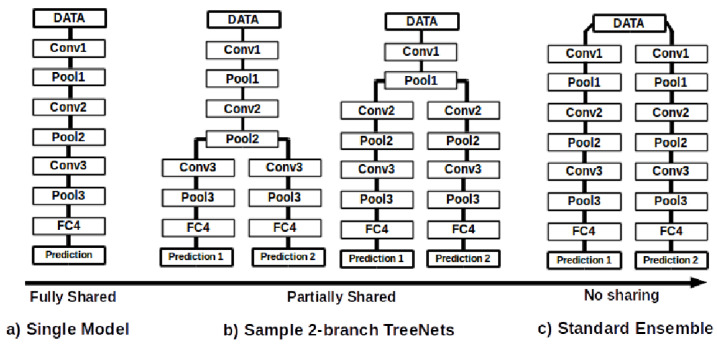
Spectrum of parameter sharing in TreeNets/HydraNets (from [69]). Depending on the degree of weight sharing, the resulting model(s) can be (**a**) a single model, where all weights are shared, (**b**) a TreeNet/HydraNet, where only some weights are shared, or (**c**) a standard ensemble, where no weights are shared.

**Figure 4 entropy-26-00634-f004:**
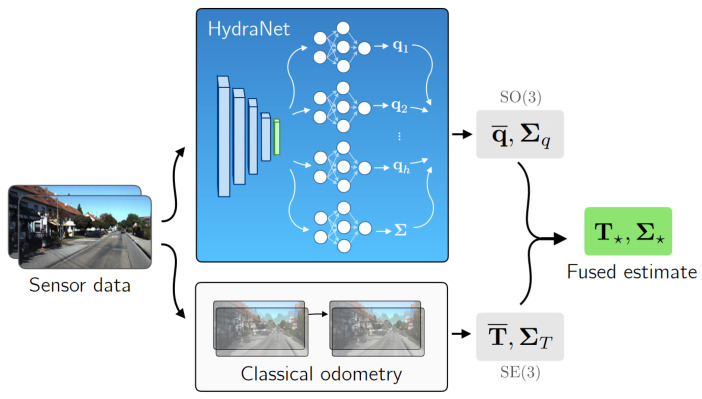
Schema of the methodology proposed by Lee et al. [69], where a HydraNet is applied to produce meaningful uncertainties, which are fused with those from classical odometry (from [23]).

**Figure 5 entropy-26-00634-f005:**
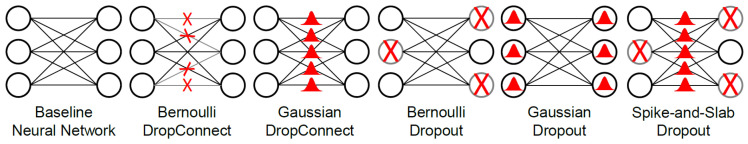
Representation of different regularisation techniques which can be used as variational distributions for the purpose of UQ. Crosses represent Bernoulli noise and bell curves represent Gaussian noise (from [86]).

**Figure 6 entropy-26-00634-f006:**
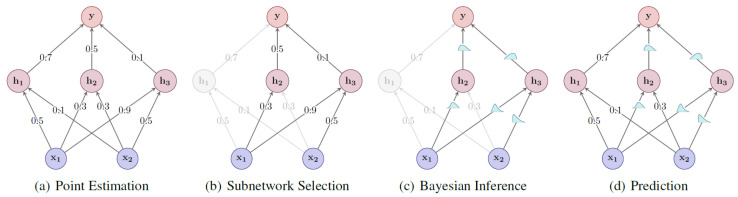
Bayesian subnetwork inference applied to a neural network in four steps (from [33]).

**Figure 7 entropy-26-00634-f007:**
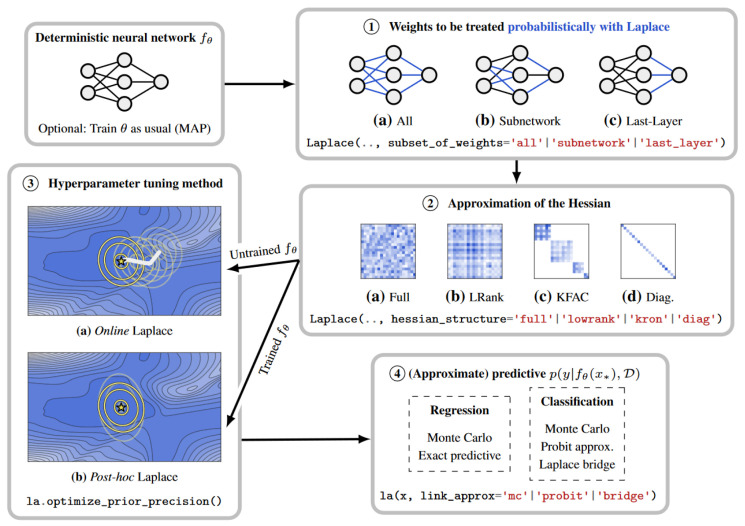
Steps to the process of adding Laplace approximation-based UQ to a neural network (from [34]).

**Figure 8 entropy-26-00634-f008:**
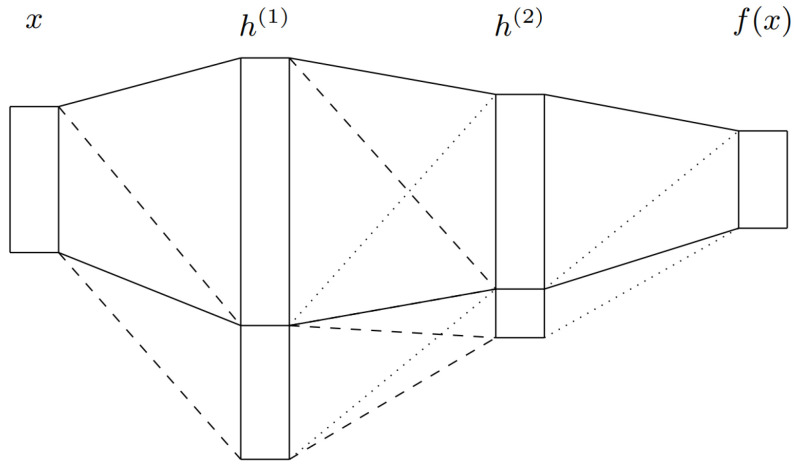
The addition of LULA units to a Neural Network (from [35]). Rectangles represent layers and solid lines represent the original MAP weights. LULA blocks are added at the bottom of the hidden layers. Dashed lines represent free parameters, while dotted lines represent zero weights.

**Figure 9 entropy-26-00634-f009:**
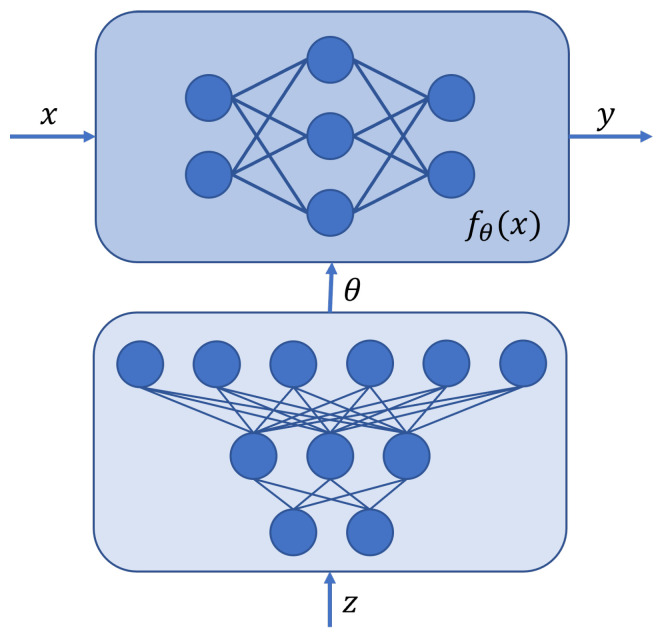
Basic architecture of a Bayesian hypernetwork, where the primary model above uses as weights the output of a secondary model. Krueger et al. [38] suggest the use of random noise as input to the latter.

**Figure 10 entropy-26-00634-f010:**
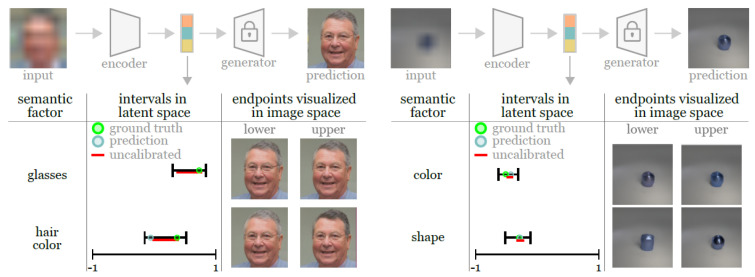
Semantically meaningful uncertainty intervals on example images sampled from the generator trained on two different datasets. The corrupt image is provided as input to the encoder which outputs a point-wise prediction and quantile predictions for each style dimension. Calibrated and uncalibrated intervals are plotted as well as their visualisations in image-space (from [40]).

**Figure 11 entropy-26-00634-f011:**
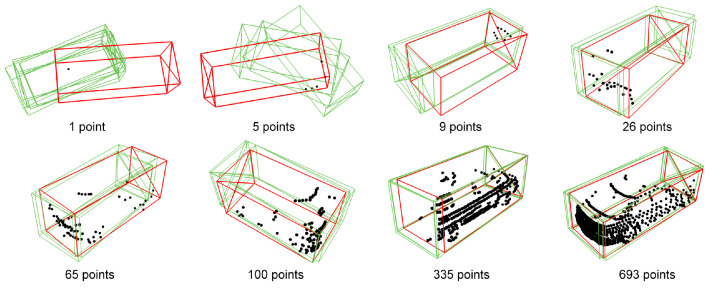
Uncertainty estimation of the annotated ground truth bounding boxes using the variance of the multiple predictions by GLENet (from [53]). The point cloud, annotated ground truth boxes and predictions of GLENet are coloured in black, red and green, respectively.

**Figure 12 entropy-26-00634-f012:**
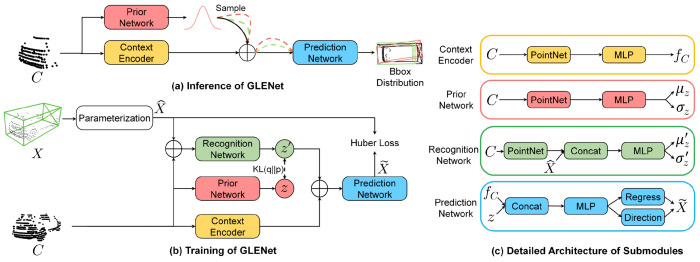
The overall workflow of GLENet (from [53]).

**Figure 13 entropy-26-00634-f013:**
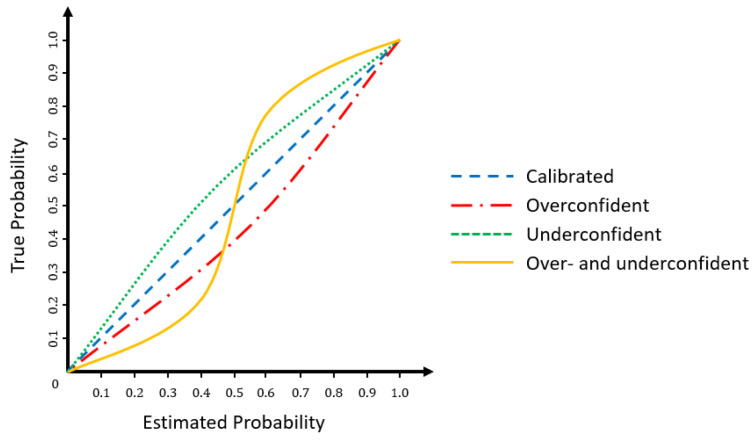
A calibration or reliability plot for classification problems shows the relationship between model estimated probabilities and the true, empirical probabilities. In well-calibrated models, the estimated probabilities should be equal to the true, empirical probabilities.

**Figure 14 entropy-26-00634-f014:**
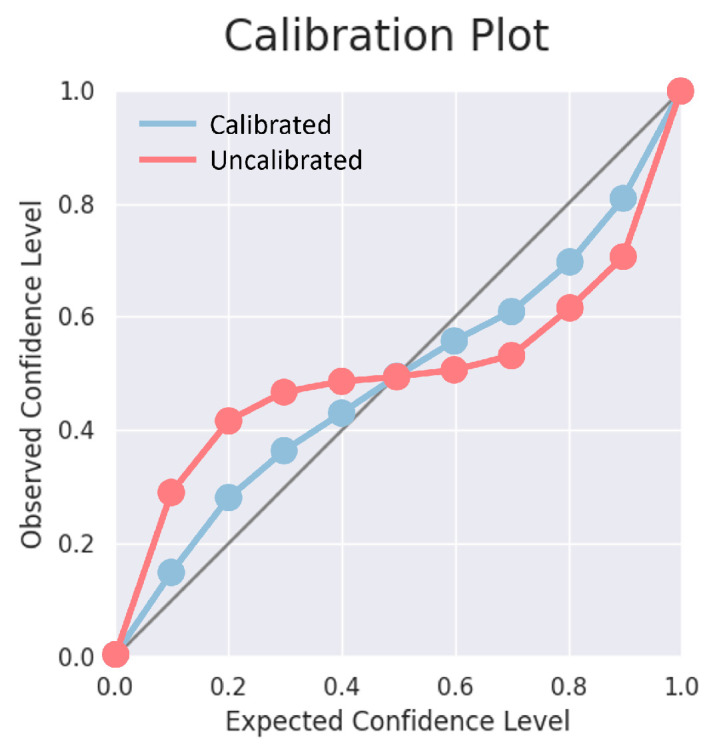
Example of a calibration plot for a regression model showing predicted and true confidence levels for an uncalibrated and a calibrated model. The grey line represents the ideal calibration.

**Figure 15 entropy-26-00634-f015:**
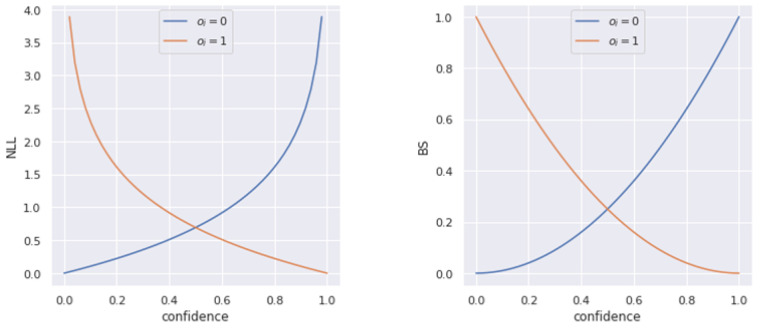
NLL (**left**) and BS (**right**) in binary classification as a function of the predicted confidence for the positive class in positive (orange) and negative samples (blue).

**Figure 16 entropy-26-00634-f016:**
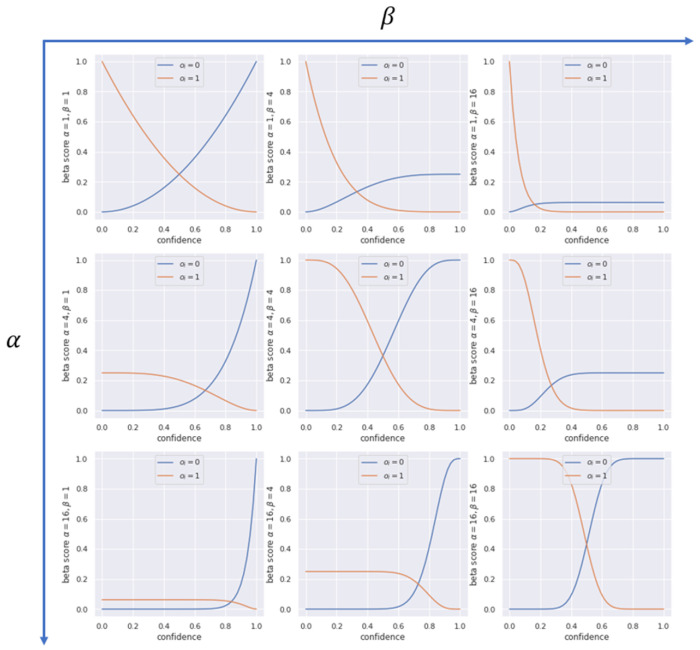
Different instantiations of the Beta Family in binary classification as a function of the predicted confidence for the positive class in positive (orange) and negative samples (blue). Different values for the parameters α and β result in different loss functions.

**Figure 17 entropy-26-00634-f017:**
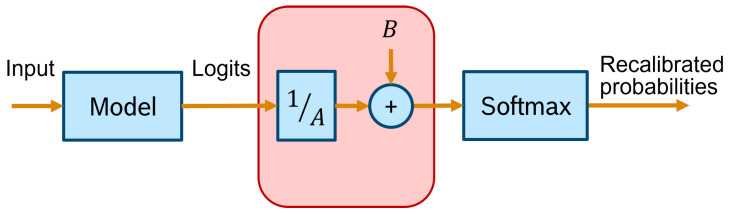
Overview of the Platt scaling method.

**Figure 18 entropy-26-00634-f018:**
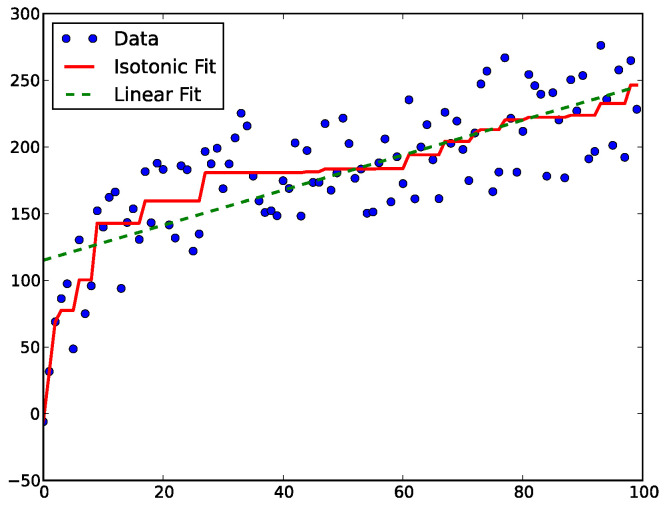
Isotonic regression versus linear regression model in fitting toy data. Image by Alexeicolin—Own work, CC BY-SA 3.0—https://commons.wikimedia.org/w/index.php?curid=23732999, accessed on 13 June 2024.

**Figure 19 entropy-26-00634-f019:**
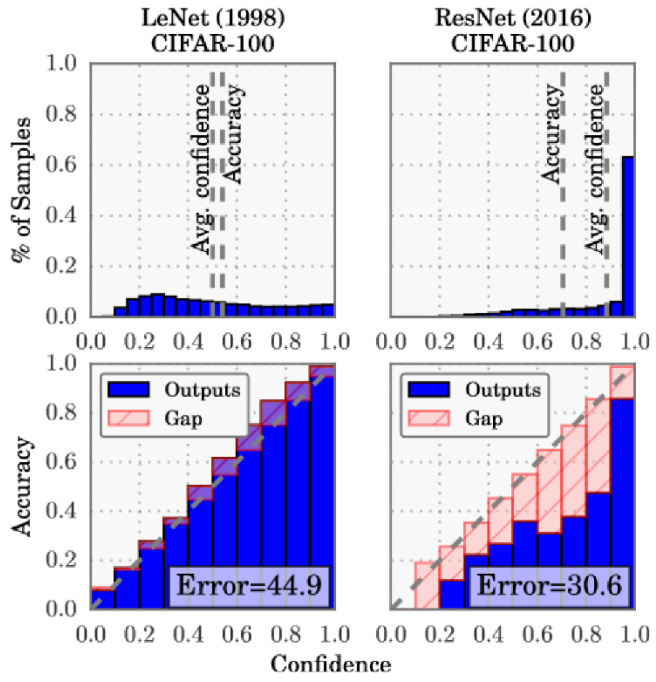
Confidence histograms (**top**) and reliability plots (**bottom**) for the older LeNet (**left**) and the more recent ResNet (right) (from [115]).

**Figure 20 entropy-26-00634-f020:**
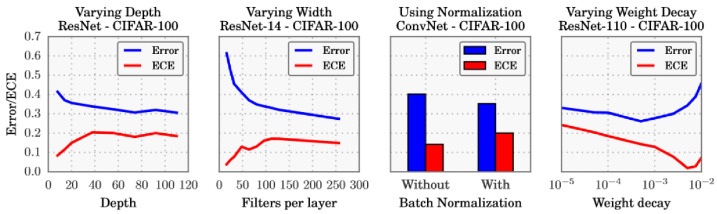
The effect of depth, width, batch normalisation, and weight decay on the classification error and the Expected Calibration Error (ECE) (from [115]).

**Figure 21 entropy-26-00634-f021:**
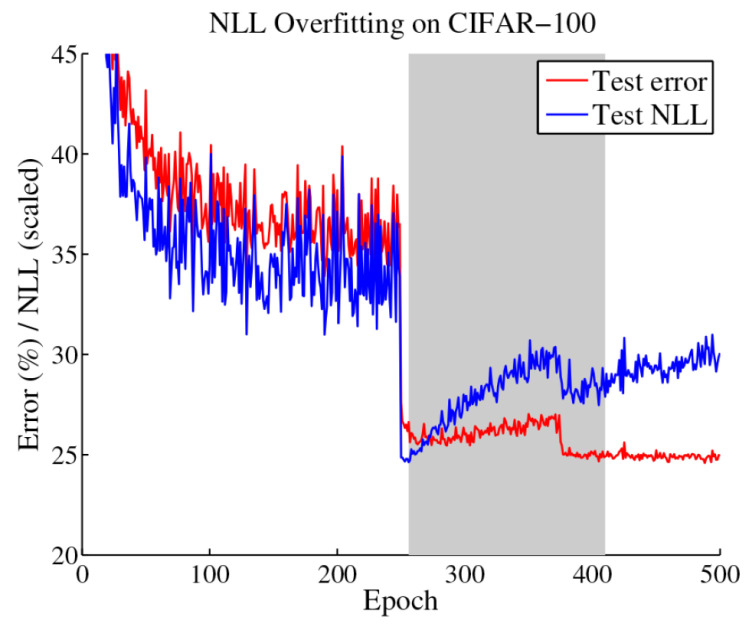
Test error and NLL of a ResNet during training (from [115]). NLL is scaled in the plot. The learning rate is dropped by a factor of 10 after 250 epochs.

**Figure 22 entropy-26-00634-f022:**
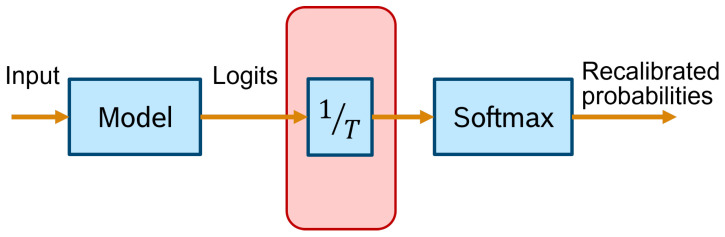
Overview of the temperature scaling method.

**Figure 23 entropy-26-00634-f023:**
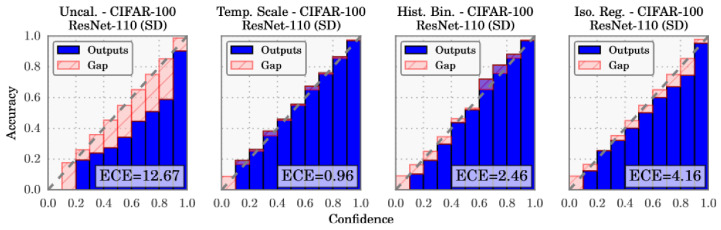
Reliability plots before and after calibration (from [115]).

**Figure 24 entropy-26-00634-f024:**
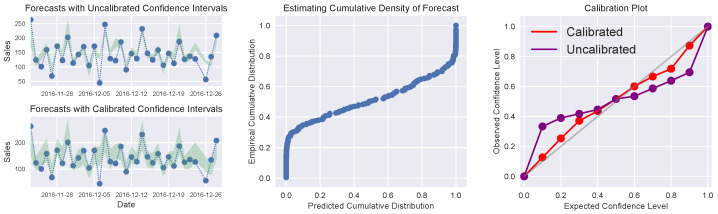
Regression calibration (from [116]). Left: Confidence intervals of probabilistic forecasts $F_{t}$ (green) before and after calibration, juxtaposed to the observed values $y_{t}$ (blue) of a time series. Middle: Scatter plot example of a recalibration dataset *D*. The number of times the target falls in a confidence interval was counted for each predicted confidence interval, i.e., $P(F_{X}\left(Y\right)\leq p)$ was computed. An isotonic regression can then be fit on these data. Right: The calibration of the estimates improves after the proposed recalibration. The plot shows the expected vs. the empirical rates of target $y_{t}$ being contained in a set of 10 confidence value bins.

**Figure 25 entropy-26-00634-f025:**
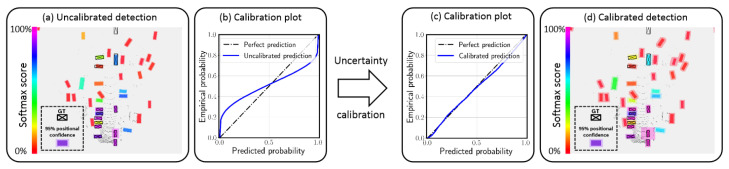
Recalibration of a SOTA LiDAR object detection model (from [121]). The calibration error is greatly reduced as the reliability diagram (**c**) demonstrates. The 95% confidence intervals for the horizontal position predictions are drawn as shaded areas for each detection.

**Figure 26 entropy-26-00634-f026:**
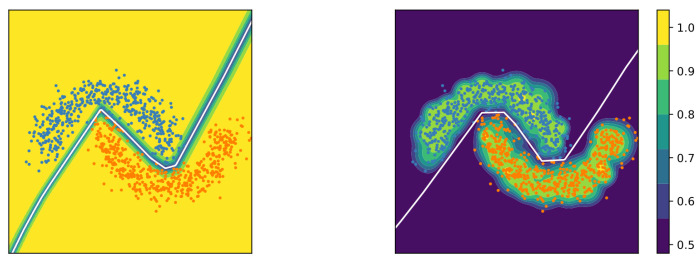
Illustration on toy dataset (from [142]): Colour-coded confidence in the prediction (yellow indicates high confidence $max_{y}\hat{p}\left(y|x\right)\approx 1$, whereas dark purple regions indicate low confidence $max_{y}\hat{p}\left(y|x\right)\approx 0.5$ for a normal neural network (**left**) and the CCU neural network (**right**). The decision boundary is shown in white for both models. The CCU model retains high confidence predictions in regions close to the training data, and roughly uniform lower confidence far from the training data. In contrast, the normal neural network is overconfident everywhere except very close to the decision boundary.

**Figure 27 entropy-26-00634-f027:**
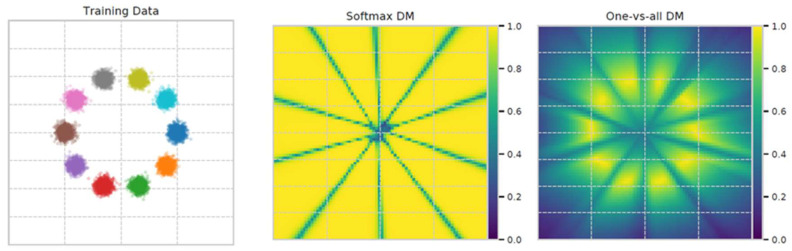
Confidence landscapes for a 2D toy example with 10 classes (**left**). A 2-layer fully connected neural network was trained with different loss functions. A distance-based loss function with the softmax normalisation resulted in infinite regions of space with uniform confidence divided by the decision boundaries (**center**). A distance-based method with one-vs-all normalisation learned to predict higher confidences centred on the training manifold (**right**) (from [144]).

**Figure 28 entropy-26-00634-f028:**
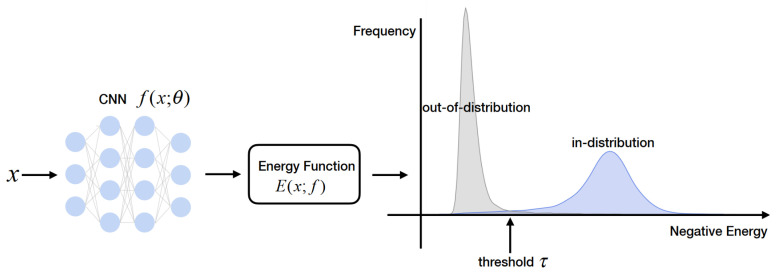
Energy-based OOD detection (from [148]).

**Figure 29 entropy-26-00634-f029:**
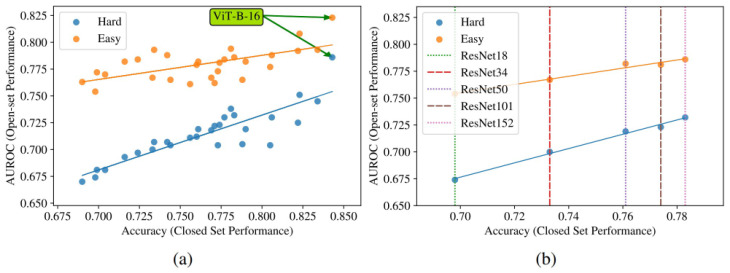
(**a**) Open-set results from different architectures on the ImageNet dataset. (**b**) ImageNet open-set results for different versions of the ResNet (from [153], image by Vaze et al. [153], licensed under CC BY 4.0 https://creativecommons.org/licenses/by/4.0/, accessed on 13 June 2024).

**Figure 30 entropy-26-00634-f030:**
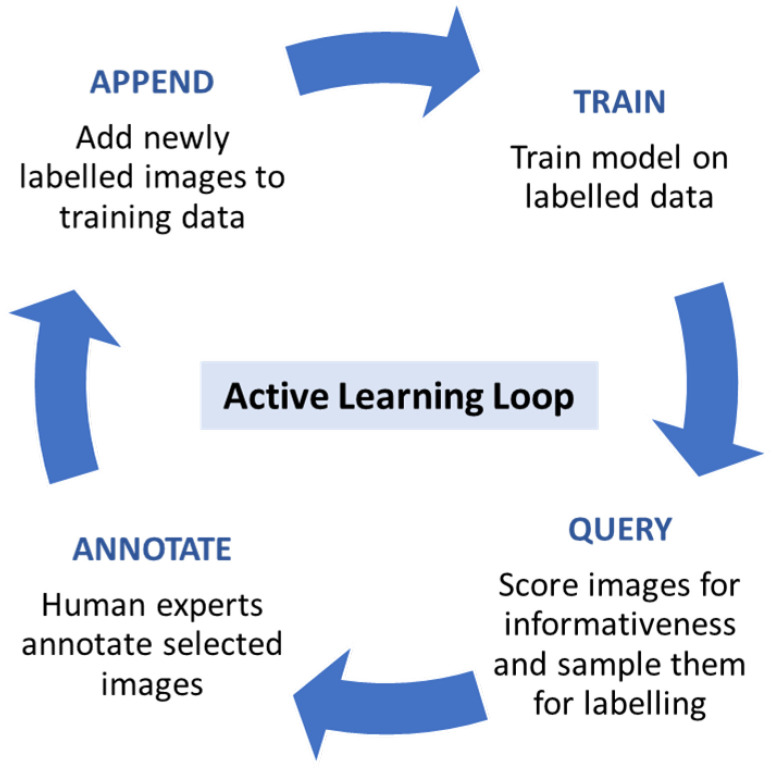
Active learning process diagram (adapted from [189]). The trained model identifies the unlabelled samples that result in higher prediction uncertainty for a new annotation iteration.

**Table 1 entropy-26-00634-t001:** Summary of the most prominent papers on UQ.

Method	Authors	Modality	AD	Task	Description
Uncertainty modelling	Kendall and Gal [18] (2017)	RGB	Y	Sem. Seg., Depth Reg.	Decomposition of uncertainty into aleatoric and epistemic components.
ADF-based uncertainty	Gast and Roth [19] (2018)	RGB	N	Class., Reg.	Model activations as normal distributions, while weights are kept as point estimates.
Test-Time Augmentations for UQ	Wang et al. [20] (2019)	RGB, Grayscale	N	Sem. Seg.	Output diversity between different augmentations at test time as aleatoric uncertainty.
Deep ensemble	Lakshminarayanan et al. [21] (2017)	RGB	N	Class.	Deep ensembles to produce uncertainty estimates.
Deep ensemble Equivalent	Ashukha et al. [22] (2020)	RGB	N	Class.	Small deep ensembles are equivalent to more sophisticated ensemble techniques.
HydraNet	Peretroukhin et al. [23] (2019)	RGB	Y	Odometry	Hydranet to produce calibrated uncertainty estimates regarding visual odometry.
Hydra Distillation	Tran et al. [24] (2020)	RGB, Tabular	N	Class., Reg.	Ensemble distillation to a HydraNet.
Monte Carlo Dropout	Gal and Ghahramani [25] (2016)	RGB, Tabular	N	Class., Reg.	Activations are dropped during inference to sample multiple predictions and compute uncertainty.
Variational Dropout	Kingma et al. [26] (2015)	RGB	N	Class.	Gaussian Dropout can be used for UQ, while the associated dropout probability can be learned.
Spike-and-slab Dropout	McClure and Kriegeskorte [27] (2017)	RGB	N	Class.	Combinations of stochastic regularisation methods can be used for UQ, with improvements in performance.
Distilled Dropout Network	Gurau et al. [28] (2018)	RGB	N	Class., Obj. Det.	Distils an MC Dropout ensemble into a single student model.
MC Dropout Approximation	Brach et al. [29] (2020)	Tabular	N	Reg.	Architecture matching 1st and 2nd moments of the MC Dropout ensemble, using moment propagation.
Bayes by Backprop	Blundell et al. [7] (2015)	RGB, Tabular	N	Class., Reg.	Backprop-compatible variational method to estimate posterior of NN parameters.
Softplus normalisation	Shridhar et al. [30] (2018)	RGB	N	Class.	Bayes by backprop where the output of a Softplus function in the final layer is normalised.
Uncertainty-guided Continual Bayesian NNs	Ebrahimi et al. [31] (2020)	RGB	N	Class.	Learning rate is adapted for individual parameters according to the uncertainty, enabling continual learning.
Scalable Laplace Approximation	Ritter et al. [32] (2018)	RGB	N	Class., OOD	Weights are modelled as a Gaussian distribution using a Laplace approximation of the loss curvature.
Subnetwork inference	Daxberger et al. [33] (2021)	RGB, Tabular	N	Class., Reg.	Apply the Bayesian treatment with Laplace approximation only to a subset of weights, to improve efficiency.
Laplace Redux Framework	Daxberger et al. [34] (2021)	RGB, Text	N	Class., OOD, Reg.	Framework for Laplace-based UQ, where an approximation of the Hessian is used.
Learnable Uncertainty under Laplace Approximations (LULA)	Kristiadi et al. [35] (2021)	RGB, Tabular	N	Class., OOD, Reg.	LULA units are added to hidden layers and improve the Laplace approximations’ performance.
Mixture Density Networks	Choi et al. [36] (2018)	Tabular	Y	Reg., Vehicle control	Mixture density networks uncertainty-aware regression in a sampling-free manner.
MDNs for Obj. Det.	Choi et al. [37] (2021)	RGB	N	Obj. Det., AL	Mixture density networks for Obj. Det. to perform active learning.
Bayesian Hypernetwork	Krueger et al. [38] (2017)	RGB, Tabular	N	Class./Reg., AL, OOD	Hypernetworks output weights for another network, inducing parameter variability.
Quantum Information Potential Field (QIPF)	Singh and Principe [39] (2021)	RGB	N	Class.	Application of perturbation theory for deterministic single-shot UQ.
Semantic uncertainty intervals	Sankaranarayanan et al. [40] (2022)	RGB	N	Generation: Super-resolution, Inpainting	Estimation of principled uncertainty intervals on the semantic latent variables.
Normalising Flow Ensembles	Berry and Meger [41] (2023)	Tabular	N	Reg., AL	Normalising Flows, which capture aleatoric uncertainty, are extended for epistemic uncertainty estimation with ensembles.
Evidential Deep Learning	Sensoy et al. [42] (2018)	RGB	N	Class.	The softmax output activation function is replaced with the ReLU, and the output is used as the parameter set of a Dirichlet distribution.
Prior Network	Malinin and Gales [43] (2018)	RGB	N	Class.	Leverages the distribution over distributions of the output to compute uncertainties, including a distributional uncertainty.
Deep Evidential Regression	Amini et al. [44] (2020)	Tabular	N	Reg.	Evidential method applied to regression using high-order evidential priors.
Probabilistic LiDAR 3D Vehicle Detector	Feng et al. [45] (2018)	LiDAR	Y	Obj. Det.	3D object detector which performs UQ, employing direct modelling and MC Dropout. Uses a Region Proposal Network to propose objects.
Spatial Dropout Sampling	Amini et al. [46] (2019)	RGB	Y	Vehicle control	Leverages spatial dropout to compute uncertainty and control wheel rotation.
Influence factors analysis	Phan et al. [47] (2019)	RGB	Y	Sem. Seg.	Analysis of effects of influence factors on the perception performance and the uncertainty measures, using synthetic data.
SalsaNext	Cortinhal et al. [48] (2020)	LiDAR	Y	Sem. Seg.	Sem. Seg. range view-based model which performs UQ, employing direct modelling and MC Dropout.
Deep ensemble for AD	Gustafsson et al. [49] (2020)	RGB	Y	Sem. Seg.,	Depth Reg. Deep ensembles to produce uncertainty estimates in the AD context.
MonoFlex	Zhang et al. [50] (2021)	RGB	Y	Reg.	Monocular 3D Obj. Det. using direct modelling to weight ensemble members.
Prediction Surface Uncertainty (PURE)	Catak et al. [51] (2021)	RGB	Y	Obj. Det.	Uncertainty-aware Obj. Det. using the area of the prediction surface as the epistemic uncertainty.
CertainNet	Gasperini et al. [52] (2021)	RGB	Y	Obj. Det.	Leverages DUQ concepts to estimate sampling-free uncertainty for all Obj. Det. parameters.
Generative Label Uncertainty Estimation (GLENet)	Zhang et al. [53] (2022)	LiDAR	Y	Obj. Det.	Improves Obj. Det. by employing location UQ based on the diversity of potentially plausible bounding boxes.
Uncertainty-aware RL	Wu et al. [54] (2023)	RGB	Y	Environment, Agent Modelling	UQ for RL applied to an Ensemble Environment Model to predict transition dynamics.

*AD*: *Y*/*N*—Autonomous Driving domain: Yes/No; *Obj. Det.*—Object Detection; *Class*.—Classification; *Reg*.—
Regression; *Sem. Seg*.—Semantic Segmentation.

**Table 2 entropy-26-00634-t002:** Summary of the most prominent papers on calibration.

Method	Authors	Modality	AD	Task	Description
Platt scaling	Platt [112] (1999)	Tabular	N	Class.	Multiplicative and additive scalars in the logits.
Histogram Binning	Zadrozny and Elkan [113] (2001)	Tabular	N	Class.	Calibrating by confidence bins.
Isotonic regression	Zadrozny and Elkan [114] (2002)	Tabular	N	Class.	Isotonic regression calibration.
Temperature Scaling	Guo et al. [115] (2017)	RGB	N	Class.	Single multiplicative scalar in the logits.
Confidence interval calibration	Kuleshov et al. [116] (2018)	Tabular, RGB	N	Reg.	Confidence interval calibration for regression.
Transferable Calibration	Wang et al. [117] (2020)	RGB, Text	N	Class.	Calibration in Domain Adaptation.
Local Temperature Scaling	Ding et al. [118] (2021)	RGB, Grayscale	N	Class., Sem. Seg.	Image and location dependent temp scaling.
Parameterised Temperature Scaling	Tomani et al. [119] (2022)	RGB	N	Class.	Auxiliary NN takes output logits and predicts an adequate temperature.
Sample-dependent Adaptive Temperature Scaling	Joy et al. [120] (2023)	RGB	N	Class.	Temperature prediction module, a learnable VAE encoder.
AD Obj. Det. Calibration	Feng et al. [121] (2019)	LiDAR	Y	Obj. Det.	Calibration of an object detector, using post-hoc methods and a calibration loss.
CalibNet	Brickwedde et al. [122] (2019)	RGB, LiDAR	Y	Reg.	CalibNet recalibrates variances and weights of each component of mixtures of gaussians.
Selective Scaling	Wang et al. [123] (2023)	RGB	Y	Sem. Seg.	Separating correct/incorrect prediction for scaling and more focusing on misprediction logit smoothing.

*AD: Y/N* —Autonomous Driving domain: Yes/No; *Obj. Det.*—Object Detection; *Class.*—Classification; *Reg.*—Regression; *Sem. Seg.*—Semantic Segmentation.

**Table 3 entropy-26-00634-t003:** Summary of the most prominent papers on OOD.

Method	Category	Authors	Modality	AD	Task	Description
Maximum Softmax	U	Hendrycks and Gimpel [141] (2017)	RGB, Grey, Text, Audio	N	Class., NLP, ASR	Baseline for OOD based on maximum softmax score.
Deep Ensembles	U	Lakshminarayanan et al. [21] (2017)	RGB	N	Class., Reg.	Simple ensemble models trained with adversarial outperform the same base model employing MC-dropout.
Predictive Uncertainty	U	Ovadia et al. [68] (2019)	RGB	N	Class.	Comparison of literature methods using predictive uncertainty.
Certified Certain Uncertainty (CCU)	U/G	Meinke and Hein [142] (2020)	RGB	N	Class.	Improve ReLU classifier using CCU, where Gaussian Mixture Models are used to controllably produce data far away from the training distribution and ensure that the used classifier reports uniform confidence for an OOD sample.
Extreme-value theory (EVT)	U/G	Mundt et al. [143] (2019)	RGB	N	Class., OSR	Combination of uncertainty with EVT and advantages of using a generative architecture for OSR in classification.
One-vs-All (OVA) Classifiers	U/G	Padhy et al. [144] (2020)	RGB	N	Class.	Binary classifier per class and distance-based logit representation to encode uncertainty as a measure of distance to the training manifold.
Shifting transformation learning	U/SS	Mohseni et al. [145] (2022)	RGB	N	Class.	Learning the optimal shifted representations of the training data to improve representation learning-based OOD detection.
Activation Shaping (ASH)	E	Djurisic et al. [146] (2023)	RGB	N	Class.	Degrade object representations and compare the impact of the network’s new output with the original one, given that representations learned for ID samples by the model are more robust than those of OOD samples.
Compounded Corruption (CnC)	S	Hebbalaguppe et al. [147] (2023)	RGB	N	Class.	OOD detector trained with a (K+1) classifier, where K are the ID classes and the additional class corresponds to the proxy OOD data (synthesised from corruption of ID images).
Energy Score	E	Liu et al. [148] (2020)	RGB	N	Class.	Propose an energy score based on the logit outputs of the network for OOD detection.
Minimum Others Score (MOS)	U/S	Huang and Li [149] (2021)	RGB	N	Class.	Group softmax is used to compute the class probabilities within each group of related classes and samples are classified as OOD when they surpass the MOS.
Rectified Activations (ReAct)	U/E	Sun et al. [150] (2021)	RGB	N	Class.	Truncation of the activations at an upper limit to attenuate the outsized activation of a few selected hidden units and improve ID/OOD distributions separability.
Dropout Bayesian Neural Networks	U	Nguyen et al. [151] (2022)	RGB, Text, PE Files	N	Img. Class., Lang. Class., Malware Det.	Measure the dispersion from randomised embedding features, with the assistance of Cosine Distance, capturing norm and angular information.
OpenDet	U	Han et al. [152] (2022)	RGB	N	Class.	Expand low-density regions, associated with unknown classes, by producing more compact features for known classes.
Semantic Shift Benchmark (SSB)	D	Vaze et al. [153] (2022)	RGB	N	Class.	Improving OSR baseline by increasing model’s accuracy in the closed-set. Present new ‘Semantic Shift Benchmark’, which divides the test set on ‘Easy’ and ‘Hard’ based on semantic similarity of the open-set categories to the training classes.
SEM	G	Yang et al. [154] (2022)	RGB	N	Class.	Propose a semantics-oriented score function for full-spectrum OOD detection, with training ID, covariate-shifted ID, near-OOD and far-OOD.
Margin Entropy loss and Separability	U	Blei et al. [155] (2022)	RGB	N	Obj. Det.	Margin Entropy loss to avoid overconfidence on OOD samples and introduce new metric called Separability.
Open Long-Tailed Recognition++ (OLTR++)	E/U/D	Liu et al. [156] (2022)	RGB	N	Class.	Increasing the robustness of long-tail cases’ detection by relating embeddings of common and rare cases.
Auxiliary GAN	U/G	Nitsch et al. [157] (2021)	RGB	Y	Class.	Auxiliary GAN to produce OOD samples during training and obtain the Gaussian distribution for each ID class.
VAE	G	Feng et al. [158] (2021)	RGB	Y	Class.	VAE to model the data distribution as gaussian features that represent ID and OOD objects.
OSOD-III	U/E	Hosoya et al. [159] (2022)	RGB	Y	OSOD, Obj. Det.	More feasible definition to OSOD, which aims only at the detection of unknown objects belonging to the same super-classes as known classes.
Optical flow OOD detector	G	Yuhas and Easwaran [160] (2022)	RGB	Y	Obj. Det.	Real-time OOD detection in a mobile robot using a YOLO object detector and an optical flow OOD detector.
OOD for LiDAR Obj. Det.	U, D, G	Huang et al. [161] (2022)	LiDAR	Y	Obj. Det.	Definition of OOD for object detection, presentation of method for OOD data augmentation, and evaluation with different OOD methods.

*AD*: *Y*/*N* —Autonomous Driving domain: Yes/No; Category: *S*—Supervised Classification, *D*—Distance-based,
*SS*—Self-Supervised, *G*—Generative, *U*—Uncertainty-based, *E*—Energy-based; (*Img*.) *Class*.—(Image) Classification;
*ASR*—Automatic Speech Recognition; *Reg*.—Regression; *PE Files*—Portable Executable Files; *Lang.
Class*.—Language Classification; *Malware Det*.—Malware Detection; *Obj. Det*.—Object Detection.

**Table 4 entropy-26-00634-t004:** Summary of the most prominent papers on AL.

Method	Authors	Modality	AD	Task	Description
GALAXY	Zhang et al. [182] (2022)	RGB	N	Class.	Graph-based active learning to automatically and adaptively select more class-balanced examples for labelling.
Balanced active learning	Jin et al. [183] (2022)	RGB	N	Class.	Identification of object similarity through the values of the embedded space of a self-supervision optimised sub-module.
Loss Prediction	Yoo and Kweon [184] (2019)	RGB	N	Class.	AL guided by a loss prediction module.
Deep active learning	Geifman and El-Yaniv [185] (2017)	RGB	N	Class.	Identification of outliers using activation outputs throughout the model.
VL4Pose	Shukla et al. [186] (2022)	RGB	N	Pose Est.	AL through the detection of OOD samples, which correspond to low likelihood outputs.
Uncertainty calibration in AL	Li and Alstrøm [187] (2020)	RGB	N	Sem. Seg.	Better model calibration contributes to improve the active learning framework, especially with region acquisition.
Active and incremental learning	Lin et al. [188] (2020)	LiDAR, ALS	N	Sem. Seg.	UQ for each region of the point-cloud to select the most informative tiles to be labelled and implementation of incremental fine-tuning strategy to keep previous knowledge.
Scalable active learning	Haussmann et al. [189] (2020)	RGB	Y	Obj. Det.	Scalable production system for AL using an ensemble of U-Nets and UQ.
Deep active learning	Feng et al. [82] (2019)	LiDAR	Y	Obj. Det.	Object classification score and 3D geometric information prediction based on 2D proposals on camera images.
Rare Example Mining	Jiang et al. [190] (2022)	LiDAR	Y	Obj. Det.	Rare example mining by estimating latent feature density estimation with a normalizing flows model.
Sequence-based active learning	Denzler et al. [191] (2022)	Infrared, RGB, LiDAR	Y	Obj. Det.	Sequence-based AL, as opposed to individual frames selection for annotation.
One Class One Click	Wang et al. [192] (2022)	LiDAR	Y	Sem. Seg.	AL at sub-cloud level.

*AD: Y/N*—Autonomous Driving domain: Yes/No; *AL*—Active Learning; *Obj. Det.*—Object Detection; *Class.*—Classification; *Sem. Seg.*—Semantic Segmentation; *ALS*—Airborne Laser Scanning; *Pose Est.*—Pose Estimation.

## Data Availability

No new data were created or analysed in this study. Data sharing is not applicable to this article.

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
