# Peer review of "The Road to Safety: A Review of Uncertainty and Applications to Autonomous Driving Perception"

_entropy, 2024, doi:10.3390/e26080634_

Round 1

Reviewer 1 Report

Comments and Suggestions for Authors

This paper presents a review of recent research on uncertainty-related autonomous driving perception tasks, specifically summarizing the approaches in uncertainty quantification, out-of-distribution detection, calibration, and UQ/OOD-related active learning.

Overall, this is a clear and thorough review of the uncertainty research related to the perception of autonomous driving. One additional set of approaches that I think is also very important in terms of uncertainty quantification and OOD and are not mentioned in this paper are the evidential deep learning methods such as deep evidential regression.

Author Response

Comments 1: "One additional set of approaches that I think is also very important in terms of uncertainty quantification and OOD and are not mentioned in this paper are the evidential deep learning methods such as deep evidential regression."

Response 1: Agree. We have, accordingly, included 3 entries in Table 1 (Section 2.3 Methods, Page 5), referring to methods with names "Evidential Deep Learning", "Prior Network", "Deep Evidential Regression". Additionally,  in Section 2.3 Methods, Page 13 Lines 370-391, two paragraphs were included, discussing precisely such methods. Please refer to the updated manuscript for confirmation of the changes.

Reviewer 2 Report

Comments and Suggestions for Authors

Summary:

The article presents a literature review of uncertainty measures, calibration methods and applications of deep learning
approaches to be applied for autonomous driving scenarios. It especially handles the difference and estimations of
various aleatoric and epistemic uncertainty measures. Additionally, some applications of uncertainty estimation are
provided for the autonomous driving use-case.

Verdict:

The article provides a very solid overview of approaches, methods, and applications of uncertainty estimation. At some points, the authors should take more care to explain the concepts, even if they are not new to the field. Generally, the discussions are sound. However, the article misses an overarching hypothesis or research questions, which the literature review tries to answer. The provided concepts are often described as qualitative. It would be interesting if a quantitative comparison is possible and how the different approaches relate. Additionally, the used mathematical formulations are taken directly from the sources, which makes them incompatible with each other. I suggest the authors try to homogenize the mathematical formulations to provide an ease of understanding to the reader. Finally, a comparison of probabilistic and entropy-based uncertainty estimation would be helpful, as both concepts are presents in the described approaches.

Details:

Page 3: 77-83: The description here is mathematically correct; however, it does not provide any explanation towards the semantic of the variables. Additionally, the second part does not describe the typical behavior of a BNN as it will generally compute the probability of classification output and not the probability of the weights, even though this is included. I propose you clarify this relation and especially explain the relation between q_\theta(w) and p(\theta|D).

Page 3: Section 2.2: There is no discussion of the properties of distance-based uncertainty in relation to the previous definitions.

Page 6: Equation 13 is a special case of Equation 14. Consequently, Eq. 13 can be omitted.
Page 6: 141-143: How can uncertainty improve the accuracy of the model?
Page 6: Equation 15 does not fit with the original definitions of aleatoric uncertainty. I think you should clarify the difference in modelling.
Page 6: 161: MC Dropout was never explained.

Author Response

Comments 1: "However, the article misses an overarching hypothesis or research questions, which the literature review tries to answer."

Response 1: Agree. In accordance with the comment, the revised manuscript now includes two explicit research questions in Section 1 Introduction, page 1, lines 24-32.

Comments 2: "The provided concepts are often described as qualitative. It would be interesting if a quantitative comparison is possible and how the different approaches relate.

Response 2: Disagree. The authors, despite agreeing with the sentiment of the comment, do not feel that the suggestion in the comment would be viable for this particular review. The array of methods examined in this document present substantial differences in benchmarking setups, metrics employed and, most importantly, purposes. This makes a systematic quantitative comparison pragmatically unfeasible. As such, no action was taken in regards to this comment.

Comments 3: "Additionally, the used mathematical formulations are taken directly from the sources, which makes them incompatible with each other. I suggest the authors try to homogenize the mathematical formulations to provide an ease of understanding to the reader. "

Response 3: Disagree. The authors understand the reasoning behind the comment, however several points arguments can be made contrary to the suggestion. First, homogenizing the mathematical formulations across different publications may inadvertently create a divergence between the original works and this review. Each publication has its unique context, objectives, and specific mathematical formulations that are tailored to address specific research questions. By homogenizing the equations, we risk losing the nuances and specificities that make each publication distinct and valuable in its own right. Secondly, the process of attempting to align them across different publications can lead to unnecessary complications. Mathematical formulations are often developed based on the specific requirements and constraints of the research problem at hand. Introducing changes to align them may result in convoluted equations that are harder to understand and interpret. This could potentially hinder the clarity and readability of the review paper and do a disservice to the reader.

Lastly, implementing such a change would require a substantial time investment. Literature reviews aim to provide a comprehensive overview and analysis of existing literature. The suggested process of homogenization would involve revisiting each publication, understanding the underlying mathematical concepts, and making adjustments accordingly. This would significantly extend the timeline for tackling the minor revisions and is not be feasible within the given timeframe.

Considering these factors, the authors believe it is more beneficial to maintain the original mathematical formulations in this literature review. This approach ensures that the unique contributions and characteristics of each paper are preserved, while also maintaining clarity and readability for the readers. As such, no action was taken in regards to this comment.

Comments 4: " Finally, a comparison of probabilistic and entropy-based uncertainty estimation would be helpful, as both concepts are presents in the described approaches."

Response 4: Disagree. The authors are of the opinion that the underlying nature of uncertainty methods is both probabilistic and based on entropy. In this light, these cannot be dissociated, hence no comparison between probabilistic and entropy-based  uncertainty estimation was provided. As such, no action was taken in regards to this comment.

Comments 5: "Page 3: 77-83: The description here is mathematically correct; however, it does not provide any explanation towards the semantic of the variables. Additionally, the second part does not describe the typical behavior of a BNN as it will generally compute the probability of classification output and not the probability of the weights, even though this is included. I propose you clarify this relation and especially explain the relation between q_\theta(w) and p(\theta|D)."

Response 5: Agree. In accordance with the comment, the authors added the highlighted clarification in Section 2.1, page 3, lines 87-91.

Comments 6: "Page 3: Section 2.2: There is no discussion of the properties of distance-based uncertainty in relation to the previous definitions."

Response 6: Agree. The highlighted paragraph was added to address this oversight. It can be found in Section 2.2, page 4, lines 125-129.

Comments 7: "Page 6: Equation 13 is a special case of Equation 14. Consequently, Eq. 13 can be omitted."

Response 7: Agree. The authors removed Equation 13, as in accordance with the comment and adapted the neighbouring text to better fit the new context. Please refer to section 2.3.1, page 6, lines 154-155.

Comments 8:  "Page 6: 141-143: How can uncertainty improve the accuracy of the model?"

Response 8: Agree. The authors see the validity of the comment and proceeded to clarify it in section 2.3.1, page 6, lines 156-157.

Comments 9: "Page 6: Equation 15 does not fit with the original definitions of aleatoric uncertainty. I think you should clarify the difference in modelling"

Response 9: Agree. In order to avoid misinterpretation coming from the divergence of modelling, the authors removed Equation 15 and changed the associated text, that can now be found in section  2.3.1, page 6, lines 173-175.

Comments 10: "Page 6: 161: MC Dropout was never explained."

Response 10: Partially agree. To avoid confusion created by this reference instance of MC Dropout, the authors removed the sentence "The proposed aleatoric uncertainty is compared and combined with epistemic uncertainty, using \ac{MC} Dropout." as it was not relevant in the context (AL). MC Dropout is, in fact, explained in Section 2.3.2 Epistemic Uncertainty Quantification, page 7, line 251.
